# Experimental validation of the free-energy principle with in vitro neural networks

**Takuya Isomura** [1] ✉, **Kiyoshi Kotani**[2], **Yasuhiko Jimbo**[3] **& Karl J. Friston** [4,5]

Empirical applications of the free-energy principle are not straightforward because they entail a commitment to a particular process theory, especially at the cellular and synaptic levels. Using a recently established reverse engineering technique, we confirm the quantitative predictions of the free-energy principle using in vitro networks of rat cortical neurons that perform causal inference. Upon receiving electrical stimuli—generated by mixing two hidden sources—neurons self-organised to selectively encode the two sources. Pharmacological up- and downregulation of network excitability disrupted the ensuing inference, consistent with changes in prior beliefs about hidden sources. As predicted, changes in effective synaptic connectivity reduced variational free energy, where the connection strengths encoded parameters of the generative model. In short, we show that variational free energy minimisation can quantitatively predict the self-organisation of neuronal networks, in terms of their responses and plasticity. These results demonstrate the applicability of the free-energy principle to in vitro neural networks and establish its predictive validity in this setting.

Elucidating the self-organising principles of biological neural networks is one of the most challenging questions in the natural sciences, and should prove useful for characterising impaired brain function and developing biologically inspired (i.e., biomimetic) artificial intelligence. According to the free-energy principle, perception, learning, and action—of all biological organisms—can be described as minimising variational free energy, as a tractable proxy for minimising the surprise (i.e., improbability) of sensory inputs[1,2]. By doing so, neuronal (and neural) networks are considered to perform variational Bayesian inference[3]. (Table 1 provides a glossary of technical terms used commonly in the free-energy principle and active inference literature). This inference follows from treating neuronal dynamics as a gradient flow on variational free energy, which can be read as a form of belief updating about the network's external milieu. The free energy in question is a function of a generative model that expresses a hypothesis about how sensory data are generated from latent or hidden states. However, to apply the free-energy principle at the cellular and synaptic levels, it is necessary to identify the requisite generative model that explains neuronal dynamics (i.e., inference) and changes in synaptic efficacy (i.e., learning).

The activity of neurons has also been modelled with realistic spiking neuron models[4–6] or reduced rate coding models[7]. Moreover, synaptic plasticity—that depends on the firing of pre- and postsynaptic neurons[8–12]—has been modelled as Hebbian-type plasticity rules[13–15]. Although a precise link between the equations that underwrite these models—derived from physiological phenomena—and the corresponding equations from the free-energy principle has not been fully established, we recently identified a formal equivalence between neural network dynamics and variational Bayesian inference[16–18]. Specifically, we reverse-engineered a class of biologically plausible cost functions—for canonical neural networks—and showed that the cost function can be cast as variational free energy, under a class of well-known partially observable Markov decision process (POMDP) models. This suggests that any (canonical) neural network, whose activity and

[1]Brain Intelligence Theory Unit, RIKEN Center for Brain Science, 2-1 Hirosawa, Wako, Saitama 351-0198, Japan. [2]Research Center for Advanced Science and Technology, The University of Tokyo, 4-6-1 Komaba, Meguro-ku, Tokyo 153-8904, Japan. [3]Department of Precision Engineering, School of Engineering, The University of Tokyo, 7-3-1 Hongo, Bunkyo-ku, Tokyo 113-8656, Japan. [4]Wellcome Centre for Human Neuroimaging, Queen Square Institute of Neurology, University College London, London WC1N 3AR, UK. [5]VERSES AI Research Lab, Los Angeles, CA 90016, USA. ✉e-mail: takuya.isomura@riken.jp

## Table 1 | Glossary of terms

| Expression | Description |
| --- | --- |
| Free-energy principle (FEP) | A principle that can be applied to perception, learning, and action in biological organisms. Technically, the FEP is a variational principle of least action that describes action and perception as, effectively, minimising prediction errors. |
| Variational Bayesian inference | An approximate Bayesian inference scheme that minimises variational free energy as a tractable proxy for—or bound on—surprise. Minimising surprise is equivalent to maximising the evidence for a generative model. In machine learning, variational free energy is known as an evidence bound. |
| Prior belief | Probabilistic beliefs about unobservable variables or states prior to receiving observations, denoted as $P(\vartheta)$. |
| (Approximate) Posterior belief | (Approximate) Bayesian belief about unobservable variables or states after receiving observations, denoted as $Q(\vartheta) \approx P(\vartheta|o)$. |
| Likelihood | The likelihood of an observation given unobservable states, denoted as $P(o|\vartheta)$. |
| Generative model | Probabilistic model that expresses how unobservable states generate observations, defined in terms of the likelihood and prior beliefs $P(o, \vartheta) = P(o|\vartheta) P(\vartheta)$. |
| Surprise | The surprisal or self-information, which scores the improbability of an observation under a generative model: defined as $-\ln P(o) = -\ln\left(\int P(o, \vartheta) d\vartheta\right)$. Here, $P(o)$ is known as the marginal likelihood or model evidence. It is called the marginal likelihood because it marginalises over the unknown causes an observation. |
| Variational free energy | An upper bound on surprise—or the negative of an evidence lower bound (ELBO)—defined as $F = E_{Q(\vartheta)}[-\ln P(o, \vartheta) + \ln Q(\vartheta)]$, where $E_{Q(\vartheta)}[\bullet]$ denotes the expectation over $Q(\vartheta)$. |
| Bayesian belief updating | The process of using observations to update a prior belief to a posterior belief. Usually, in biomimetic schemes, belief updating uses variational Bayesian inference, where neuronal dynamics perform a gradient descent on variational free energy. |
| Partially observable Markov decision process (POMDP) | A generic generative model that expresses unknown causes of observations in terms of discrete state spaces and categorical distributions. |

plasticity minimise a common cost function, implicitly performs variational Bayesian inference and learning about external states. This 'reverse engineering' approach—guaranteed by formal equivalence—allows us, for the first time, to identify the implicit generative model from empirical neuronal activity. Further, it can precisely link quantities in biological neuronal networks with those in variational Bayesian inference. This enables an experimental validation of the free-energy principle, when applied to these kinds of canonical networks.

Having said this, the free-energy principle is sometimes considered to be experimentally irrefutable in the sense that it can describe any observed biological data[19]. However, when applying the free-energy principle to a particular system, one can examine its predictive validity by asking whether it can predict systemic responses[18]. This offers a formal avenue for validation and application of the free-energy principle. To establish predictive validity, one needs to monitor the long-term self-organisation of neuronal networks and compare their dynamics and architecture with theoretical predictions.

To pursue this kind of validation, we used a previously established microelectrode array (MEA) cell culture system for the long-term monitoring of the self-organisation of in vitro neural networks[20,21]. We have used this setup to investigate causal inference in cortical cells obtained from rat embryos[22,23]. Causal inference is a simple form of Bayesian inference; namely, inferring and disentangling multiple causes of sensory inputs in the sense of blind source separation[24–26]. Although blind source separation is essential to explain the cocktail party effect—the ability of partygoers to distinguish the speech of one speaker from others in a noisy room[27,28]—its precise neuronal mechanisms have yet to be elucidated. We previously demonstrated that, upon receiving sensory stimuli, some populations of neurons in in vitro neural networks self-organised (or learned) to infer hidden sources by responding specifically to distinct causes[22]. Subsequently, we showed that this sensory learning is consistent with variational free energy minimisation under a POMDP generative model[23]. These results—and related in vitro work[29–35]—speak to the tractability and stability of this neuronal system, making it an ideal tool for examining theoretical predictions in a precise and quantitative fashion.

In the present work, we attempted an experimental validation of the free-energy principle by showing that it predicts the quantitative self-organisation of in vitro neural networks using an established in vitro causal inference paradigm. Henceforth, we will refer to in vitro neural networks as *neuronal networks* and reserve the term *neural network* for an in silico model. We reverse-engineered an implicit generative model (including prior beliefs), under which a neuronal network operates. We subsequently demonstrated that the free-energy principle can predict the trajectory of synaptic strengths (i.e., learning curve) as well as neuronal responses after learning, based exclusively on empirical neuronal responses at the beginning of training.

Using pharmacological manipulations, we further examined whether the change in baseline excitability of in vitro networks was consistent with the change in prior beliefs about hidden states (i.e., the state prior), confirming that priors over hidden states are encoded by firing thresholds. These results demonstrate that the self-organisation of *neuronal networks* can be cast as Bayesian belief updating. This endorses the plausibility of the free-energy principle as an account of self-organisation in neural and neuronal networks. We conclude by discussing possible extensions of our reverse engineering approach to in vivo data.

## Results

### Equivalence between canonical neural networks and variational Bayes

First, we summarise the mathematical (or natural) equivalence between canonical neural networks and variational Bayesian inference, which enables one to apply the free-energy principle to predict empirical data. In this work, we adopted an experimental setup that could be formulated as a simple POMDP generative process that does not exhibit state transitions (Fig. 1a). Here, two binary hidden sources $s = (s_1, s_2)^\mathsf{T}$ were sampled at random from a prior categorical distribution $D = (D_1, D_0)^\mathsf{T}$ in a mutually independent manner, where $D_1$ and $D_0$ are prior expectations that satisfy $D_1 + D_0 = 1$. Then, 32 sensory inputs $o = (o_1, \ldots, o_{32})^\mathsf{T}$ were generated from $s$ with a categorical distribution characterised by a mixing matrix $A$. Each element of $s$ and $o$ took either a 1 (ON) or a 0 (OFF) state. The left stimuli group $(o_1, \ldots, o_{16})$ in Fig. 1a (left) took the value of source 1 with a 75% probability or the value of source 2 with a 25% probability. In contrast, the right group $(o_{17}, \ldots, o_{32})$ took the value of source 1 or 2 with a 25% or 75% probability, respectively. Analogous to the cocktail party effect[27,28], this setup is formally homologous to the task of distinguishing the voices of speakers 1 ($s_1$) and 2 ($s_2$) based exclusively on

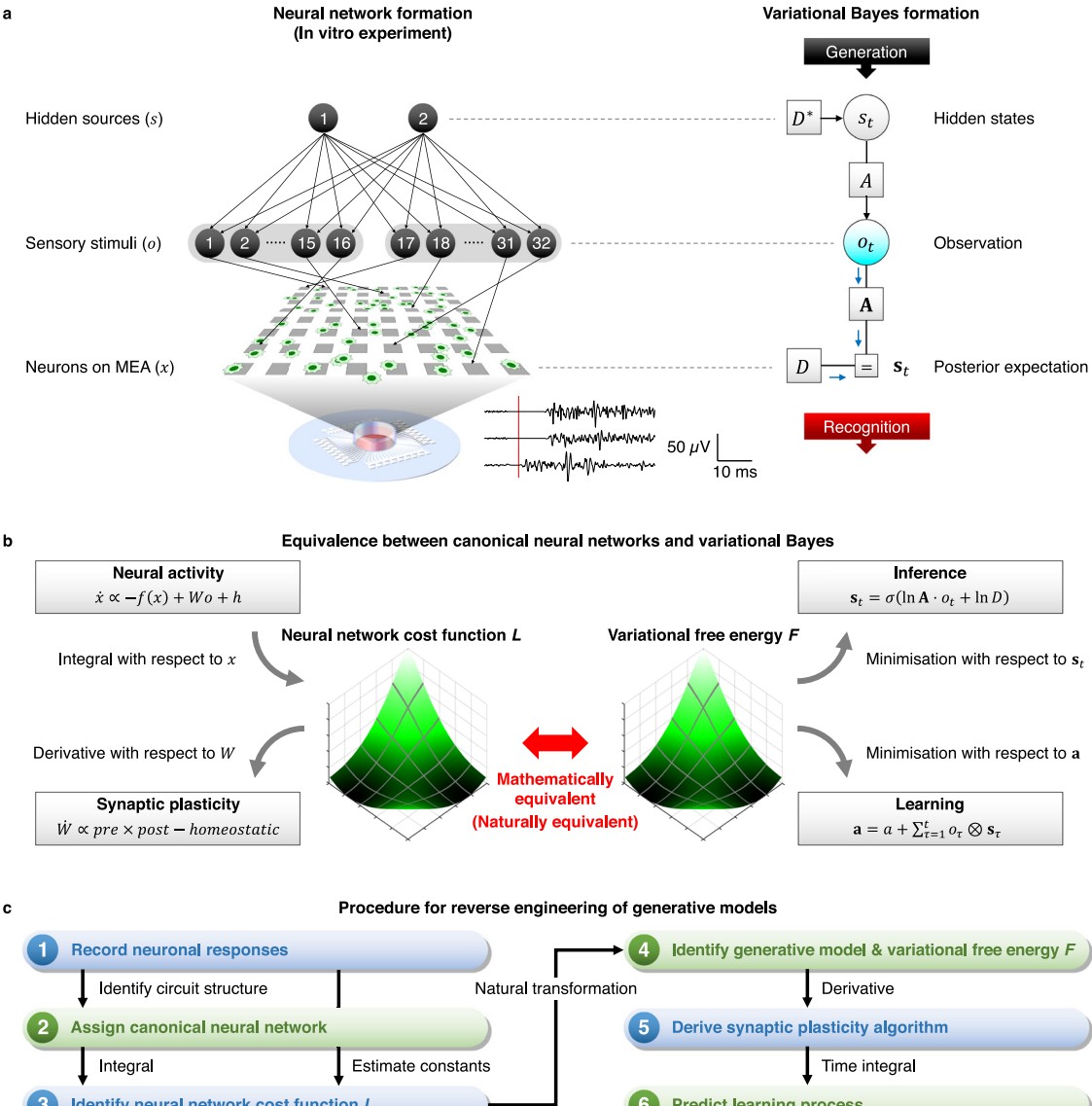

**Fig. 1 | Reverse engineering of the generative model from empirical data.** In **a**–**c**, panels on the left-hand side depict neural (and neuronal) network formation, while panels on the right-hand side depict variational Bayes formation. **a** Schematics of the experimental setup (left) and corresponding POMDP generative model (right). Two sequences of independent binary hidden sources generate 32 sensory stimuli through a mixing matrix $A$, which were applied into cultured neurons on an MEA as electrical pulses. Waveforms at the bottom represent the spiking responses to a sensory stimulus (red line). The diagram on the right-hand side depicts the POMDP scheme expressed as a Forney factor graph[67–69]. The variables in bold (e.g., $\mathbf{s}_t$) denote the posterior beliefs about the corresponding variables in non-bold italics (e.g., $s_t$). **b** Equivalence between canonical neural networks and variational Bayesian inference. See the main text and Methods for

details. **c** Procedure for reverse engineering the implicit generative model and predicting subsequent data. (1) The neuronal responses are recorded, and (2) the canonical neural network (rate coding model) is used to explain the empirical responses. (3) The dynamics of the canonical neural network can be cast as the gradient descent on a cost function. Thus, the original cost function $L$ can be reconstructed by taking the integral of the network's neural activity equation. Free parameters $\phi$ are estimated from the mean response to characterise $L$. (4) Identification of an implicit generative model and the ensuing variational free energy $F$ using the equivalence of functional forms in Table 2. (5) The synaptic plasticity rule is derived as a gradient descent on variational free energy. (6) The obtained plasticity scheme is used to predict self-organisation of *neuronal networks*. The details are provided in Methods and have been described previously[16–18].

mixed auditory inputs ($o$), and in the absence of supervision. Here, the mixing (a.k.a., likelihood) matrix ($A$) determines the mixing of the two voices, and the prior ($D$) corresponds to the frequency or probability of each speaker generating speech. Hence, neurons must unmix sensory inputs into hidden sources to perceive the underlying causes. Please refer to the Methods section 'Generative process' for the formal expression in terms of probability distributions.

In this work, we considered that in vitro neurons can be modelled as a canonical neural network comprising a single feed-forward layer of rate coding models (Fig. 1b, top left)[16]. We considered two distinct ensembles of neurons. Upon receiving sensory inputs $o$, these neurons

computed the weighted sum of sensory inputs weighted by a synaptic strength matrix $W$ to generate a response (firing intensity) $x = (x_1, x_2)^\mathsf{T}$. This canonical neural network has a certain biological plausibility because it derives from realistic neuron models[4–6] through some approximations[17]; further, its fixed point equips the rate coding model[7] with the widely used sigmoid activation function, also known as a neurometric function[36]. We will show below that this canonical neural network is a plausible computational architecture for *neuronal networks* that receive sensory stimuli.

Previous work has identified a class of biologically plausible cost functions for canonical neural networks that underlie both neuronal

responses and synaptic plasticity[16,17]. This cost function can be obtained by simply calculating the integral of the neural activity equation (Fig. 1b, middle left; see the Methods section 'Canonical neural networks' for details). The reconstructed neural network cost function $L$ is biologically plausible because both neuronal responses and synaptic plasticity equations can be derived as a gradient descent on $L$. The ensuing synaptic plasticity rule has a biologically plausible form, comprising Hebbian plasticity[13], accompanied by an activity-dependent homeostatic plasticity[37] (Fig. 1b, bottom left).

Variational Bayesian inference casts belief updating as revising a prior belief to the corresponding (approximate) posterior belief based on a sequence of observations. The experimental setup considered here is expressed as a POMDP generative model[38,39]. The inversion of this model—via a gradient descent on variational free energy—corresponds to inference. In other words, the generative model generates sensory consequences from hidden causes (i.e., two sources), while model inversion (i.e., inference) maps from sensory consequences to hidden causes (Fig. 1a, right). Variational free energy $F$ is specified by the sensory input and probabilistic beliefs about hidden states under a generative model. Minimisation of variational free energy, with respect to these beliefs, yields the posterior over hidden states $s_t$ (Fig. 1b, top right) and parameters $A$ (Fig. 1b, bottom right), realising Bayesian inference and learning, respectively. The explicit forms of posterior beliefs are described in the Methods section 'Variational Bayesian inference'.

Crucially, previous work has shown that the neural network cost function $L$ can be read as variational free energy $F$[16,17]. This equivalence allows us to identify the physiological implementation of variational Bayesian inference by establishing a one-to-one mapping between neural network quantities and the quantities in Bayesian inference, as summarised in Table 2. Namely, neural activity ($x$) of the canonical neural networks corresponds to the posterior expectation about the hidden states ($s$), synaptic strengths ($W$) correspond to the posterior expectation about the parameters ($A$), and firing threshold factors ($\phi$) correspond to the initial state prior ($D$). These mappings establish a formal relationship between a neural network formulation (Fig. 1b, left) and a variational Bayesian formulation (Fig. 1b, right). In summary, the neural activity and plasticity of canonical networks that minimise a common cost function perform variational Bayesian inference and learning, respectively.

This notion is essential because, by observing neuronal responses, we can reverse engineer the implicit generative model—under which the neuronal network operates—from empirical neuronal responses, to characterise the neuronal network in terms of Bayesian inference (Fig. 1c)[18]. Perhaps surprisingly, using the reverse engineering technique, if one can derive the neural activity equation from experimental data (Fig. 1c, steps 1,2), it is possible to identify the generative model that the biological system effectively employs (steps 3,4). This allows one to link empirical data to quantities in variational Bayesian

inference. Subsequently, by computing the derivative of variational free energy under the generative model, one can derive the synaptic plasticity predicted theoretically (step 5). In short, if one has initial neuronal response data, one can predict how synaptic plasticity will unfold over time. This means that if the free-energy principle applies, it will predict the self-organisation of neuronal networks (step 6).

The virtue of the free-energy principle is that it lends an explainability to neuronal network dynamics and architectures, in terms of variational Bayesian inference. Given this generative model, the free-energy principle provides qualitative predictions of the dynamics and self-organisation of neuronal networks, under the given experimental environment. In other words, because neuronal responses and synaptic plasticity are expected to minimise variational free energy by exploiting the shortest path (i.e., a geodesic or path of least action) on the free energy landscape, this property in turn enables us to theoretically predict a plausible synaptic trajectory (i.e., activity-dependent plasticity).

In the remainder of this paper, we examine the plausibility of variational free energy minimisation as the mechanism underlying the self-organisation of *neuronal networks*. We will compare the empirical encoding of the sources of sensory inputs with a synthetic simulation of ideal Bayesian encoding, and investigate whether variational free energy minimisation can predict the neuronal responses and plasticity of in vitro networks.

## Consistency between in vitro neural networks and variational Bayes

In this section, we verify some qualitative predictions of the free-energy principle when applied to our in vitro neural networks in terms of response selectivity (i.e., inference), plasticity (i.e., learning), and effects of pharmacological manipulations on inference and subsequent learning. Using our in vitro experimental setup[20,21], cortical cells obtained from rat embryos were cultured on an MEA dish with 64 microelectrodes on its floor (Fig. 1a, left). Each electrode was used to deliver electrical stimuli and record the spiking response. After approximately 10 days in culture, the neurons self-organised into a network and exhibited spontaneous activity, with clear evoked responses to electrical stimuli. Neurons were stimulated with the above-constructed patterns of sensory inputs (see the preceding section), comprising 32 binary sensory inputs ($o$) that were generated from two sequences of independent binary hidden sources ($s$) in the manner of the POMDP generative model above (Fig. 1a, right). When a sensory input took the value of 1, an electrical pulse was delivered to the cultured neurons. The 32 stimulation electrodes were randomly distributed over $8 \times 8$ MEAs in advance and fixed over training. Evoked extracellular activity (i.e., the early neuronal response) was recorded from 64 MEA electrodes. Each session lasted 256 s, in which a 256-time-step sequence of random stimulations was delivered every second, followed by a 244-s resting period. The training comprised

## Table 2 | Correspondence of variables and functions

| Neural network formation | | Variational Bayes formation |
|---|---|---|
| Neural network cost function | $L \Longleftrightarrow F$ | Variational free energy |
| Sensory stimuli | $o_t \Longleftrightarrow o_t$ | Observations |
| Neural response | $\begin{pmatrix} x_t \\ \overline{x_t} \end{pmatrix} \Longleftrightarrow s_t$ | State posterior |
| Synaptic strengths | $W_l \Longleftrightarrow \mathrm{sig}^{-1}(A_{1l})$ | Parameter posterior |
| Threshold factor | $\phi := \begin{pmatrix} \phi_1 \\ \phi_0 \end{pmatrix} \Longleftrightarrow \ln D$ | State prior |
| Firing threshold | $h_l = \ln \overline{\widehat{W}_l} \vec{1} + \phi_l \Longleftrightarrow \ln A_{0l} \cdot \vec{1} + \ln D_l$ | |
| Initial synaptic strengths | $\lambda_l^W \odot \widehat{W}_l^{\mathrm{init}} \Longleftrightarrow a_{1l}$ | Parameter prior |

Bold case variables (e.g., $s_t$) denote the posterior expectations of the corresponding italic case random variables (e.g., $s_t$); $\widehat{W}_l := \mathrm{sig}(W_l)$ is the sigmoid function of $W_l$ in the elementwise sense ($l = 0,1$); $W_l^{\mathrm{init}}$ is the initial value of $W_l$; and $\lambda_l^W$ is the inverse learning rate factor that expresses the insensitivity of synaptic strengths to plasticity. Please refer to previous work[16,17] for details.

100 sessions, each of which was an identical repetition of the 256 s-long random sequence.

Upon electrical simulation—generated by the mixture of the two hidden sources—our previous work showed the emergence of selective neuronal responses to either of the two sources[22,23]. Response intensity was defined as the number of spikes 10–30 ms after a stimulation (Fig. 2a) following the previous treatment[22] (see Supplementary Fig. 1a for other electrodes). This is because a large number of spikes—induced by synaptic input—were observed during that period, while most directly evoked action potentials (which were not the subject of our analyses) occur within 10 ms after stimulation[40]. The recorded neuronal responses were categorised into source 1- and source 2-preferring and no-preference groups, depending on the average response intensity, conditioned upon the hidden source (Fig. 2b). Note that each electrode can record spiking responses from one or more neurons. Learning was quantified as the emergence of functional specialisation for recognising particular sources. The response intensity of the source 1-preferring neurons changed during the training period to exhibit a strong response selectivity to source 1 (Fig. 2c, left). These neurons self-organised to fire at a high level when source 1 was ON, but had a low response rate when source 1 was OFF. Similarly, source 2-preferring neurons self-organised to respond selectively to source 2 during training (Fig. 2c, right). These changes were inhibited by N-methyl-D-aspartate (NMDA) receptor antagonist, 2-amino-5-phosphonopentanoic acid (APV) to a certain degree (Fig. 2d), indicating that the observed self-organisation depends on NMDA-receptor-dependent plasticity. These results indicate the occurrence of blind source separation at a cellular level—through activity-dependent synaptic plasticity—supporting the theoretical notion that neural activity encodes the posterior belief (i.e., expectation) about hidden sources or states[1,2].

Given the consistency between source-preferring neuronal responses and state posterior, one can then ask about the neuronal substrates for other quantities in variational Bayesian inference. In light of the above, we modelled *neuronal networks* using a canonical *neural network*, comprising a single feed-forward layer (Fig. 1b, top left). As noted above, this neural network acts as an ideal Bayesian observer, exhibiting Bayesian belief updating under a POMDP generative model (Fig. 1b, top right), where the firing threshold encodes a prior over initial states (Table 2)[16,17]. Thus, this in silico model can learn to detect hidden sources successfully when the implicit state prior matches that of the true generative process (in this case, $D_1 = 0.5$; Fig. 2e, middle). Conversely, both upregulation ($D_1 = 0.8$; Fig. 2e, right) and downregulation ($D_1 = 0.2$; Fig. 2e, left) of the state prior significantly disrupted this sensory learning (Fig. 2f). These simulations used the same empirical stimuli applied to neuronal networks. Hence, if this canonical neural network is an apt model for *neuronal networks*, the firing threshold (i.e., baseline excitability) of the *neuronal network* should encode the state prior, and changes in baseline excitability should disrupt the inference and ensuing sensory learning.

To examine this hypothesis, we asked whether pharmacological modulations of the baseline excitability of in vitro networks induce the same disruptions of inference as the alterations in the state prior in the in silico network. Pharmacological downregulation of gamma-aminobutyric acid (GABA)-ergic inputs (using a $GABA_A$-receptor antagonist, bicuculline) or its upregulation (using a benzodiazepine receptor agonist, diazepam) altered the baseline excitability of *neuronal networks*. These substances were added to the culture medium before the training period and were therefore present over training. Average response levels were higher in bicuculline-treated cultures than in control cultures. Conversely, diazepam-treated cultures exhibited lower response levels, but retained sufficient responsiveness to analyse response specificity. Crucially, alterations in neuronal responses—and subsequent learning—were observed when we pharmacologically modulated the GABAergic input level (Fig. 2g). We

observed that both hyper-excitability (Fig. 2g, right) and hypo-excitability (Fig. 2g, left) significantly suppressed the emergence of response specificity at the cellular level (Fig. 2h). This disruption of learning was observed both for source 1- and 2-preferring neuronal responses.

Effective synaptic connectivity analysis suggested that a certain amount of plasticity occurred even in the presence of bicuculline or diazepam (Supplementary Fig. 1b). The difference was observed in the specificity of connectivity emerging during the training period (Supplementary Fig. 1c). Here, the specificity was characterised with a gap in the contribution of a sensory electrode to sources 1- and 2-preferring units. While the specificity increased in all groups, it was significantly inhibited in the presence of bicuculline or diazepam.

Remarkably, our in silico model—under ideal Bayesian assumptions—could predict the effects of this GABAergic modulation on learning using a simple manipulation of the prior belief about hidden states (please compare Fig. 2e, f with Fig. 2g, h). This involved setting the prior expectations so that sensory causes were generally present (analogous to the GABAergic antagonist effect) or generally absent (analogous to the agonist effect). Physiologically, this corresponds to increasing and reducing the response intensity, respectively, which is consistent with the effects of these pharmacological manipulations on baseline activity. In terms of inference, this manipulation essentially prepares the network to expect the presence or absence of an object (i.e., a hidden source) prior to receiving sensory evidence. The key notion here is that this simple manipulation was sufficient to account for the failure of inference and subsequent learning, as evidenced by the absence of functional specialisation. Thus, changes in the prior (neuronal) representations of states provide a sufficient explanation for aberrant learning.

In summary, the emergence of response specificity observed under normal network excitability was disrupted by pharmacologically induced hyper- or hypo-excitability of the network. The canonical neural network (i.e., ideal Bayesian observer) predicted these empirical effects—of the agonist and antagonist—by reproducing the hypo-excitability (diazepam) condition, analogous to the prior belief that sources are OFF ('nothing there'), or by the hyper-excitability (bicuculline) condition, analogous to the prior belief that sources are present (ON). In either case, in vitro and in silico networks failed to perform causal inference, supporting our claim that the failure can be attributed to a biased state prior, under which they operated. These results corroborate the theoretical prediction that the firing threshold is the neuronal substrate of the state prior[16,17], validating the proposed equivalence at the cellular level. This further licences an interpretation of *neuronal network dynamics* in terms of Bayesian inference and learning.

## The free-energy principle predicts learning in neuronal networks

In this section, we examine the predictive validity of the free-energy principle by asking whether its application to *neuronal networks* can predict their self-organisation. We considered that the neuronal responses of source 1- and source 2-encoding ensembles in each in vitro networks are represented by their averaged response intensity and refer to them as $x_1$ and $x_2$, where the offset was subtracted, and the value was normalised in the range between 0 and 1. We then modelled the neuronal responses of in vitro networks in the form of a canonical neural network and estimated the requisite synaptic strengths $W$ (i.e., effective synaptic connectivity) by fitting empirical neuronal responses to the model (Fig. 3a; see the Methods section 'Reverse engineering of generative models' for details). Using these estimates, we depicted the trajectories (i.e., learning curves) evinced by subsequent neuronal responses.

First, we computed the synaptic strengths $W$ that minimised the neural network cost function $L$ using neuronal responses $x$. This

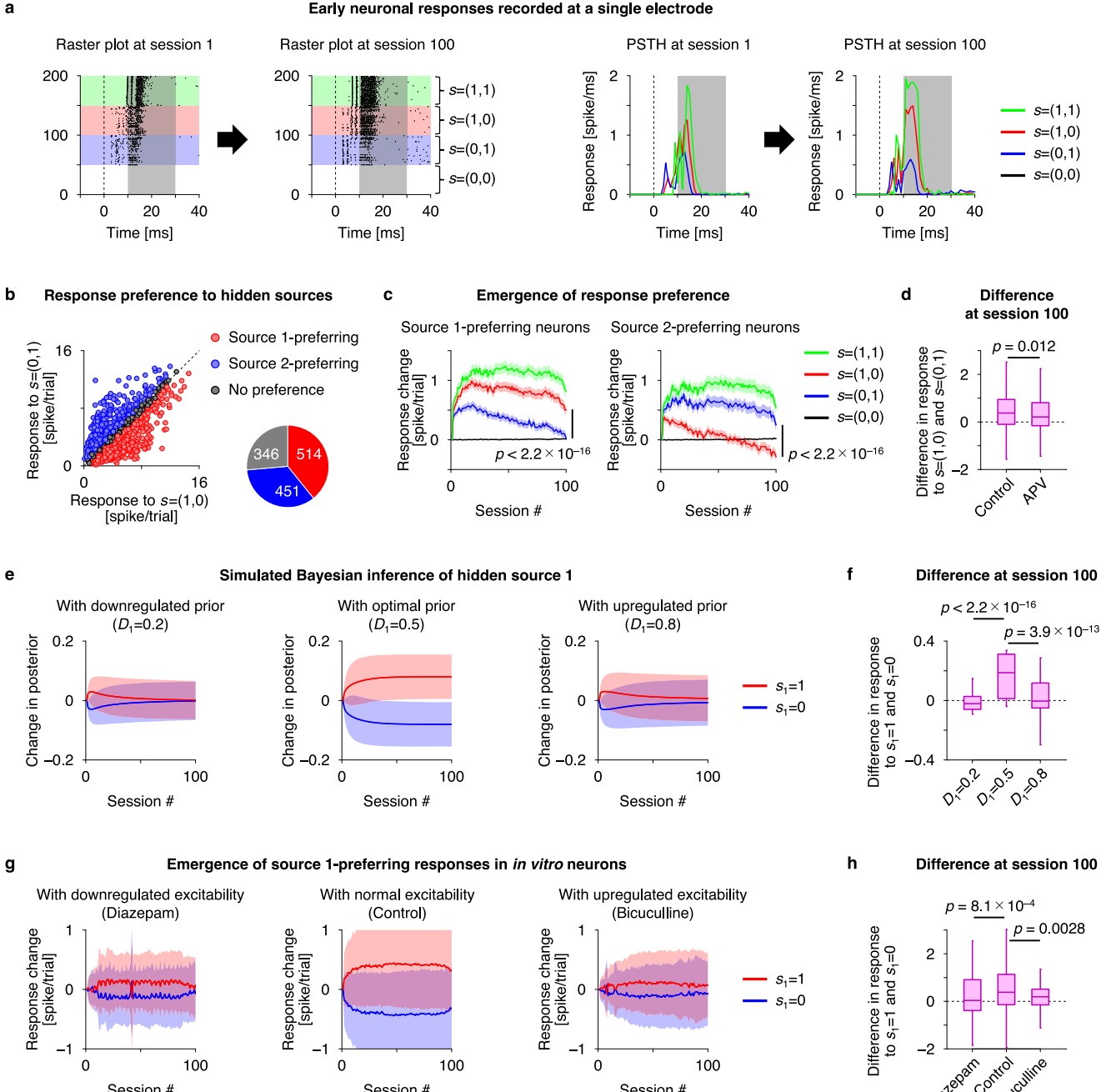

**Fig. 2 | Neuronal networks perform blind source separation, consistent with Bayesian belief updating. a** Early evoked responses of in vitro neurons recorded at a single electrode, showing a source 1-preferring neuronal response. Raster plot of spiking responses (left) and peristimulus time histogram (PSTH, right) before and after training are shown. The two sources provide four hidden state patterns, $s_t = (1,1), (1,0), (0,1), (0,0)$, and responses in these four conditions are plotted in green, red, blue, and black, respectively. Responses in shaded areas (10–30 ms after a stimulus) were used for analyses. **b** Recorded neuronal responses were categorised into source 1-preferring (red), source 2-preferring (blue), and no-preference (grey) groups. The Pie-chart indicates numbers (electrodes) in each group, obtained from 30 independent experiments. **c** Changes in evoked responses of source 1- (left) and source 2- (right) preferring neurons, respectively. Response change from session 1 is shown. Lines and shaded areas represent mean values +/− standard errors. Here and throughout, the two-sided Wilcoxon signed-rank test was used for paired comparisons. **d** Comparison of response specificities in control ($n = 965$ electrodes) and APV-treated ($n = 296$ electrodes from 9 independent experiments) culture groups. Here and throughout, the two-sided Mann–Whitney $U$ test was used for unpaired comparisons. Box-and-whisker plots in (**d**)(**f**)(**h**) follow

standard conventions: the central line indicates the median, the bottom and top box edges indicate the first and third quartiles, respectively, and the whiskers extend to the furthest data point within 1.5 times the interquartile range of the first or third quartile. **e** Simulations of ideal Bayesian observers. The posterior belief about source 1 with varying hidden state priors is shown. Red and blue lines represent how much the posterior expectation changes, when source 1 is ON or OFF, respectively ($n = 100$ simulations for each condition). In (**e**) (**g**), changes in response from session 1 were computed and then the averaged response (trend) in each session was subtracted to focus on response specificity to the preferred source. Lines and shaded areas in (**e**) (**g**) represent mean values +/− standard deviations. **f** Difference in responses to to $s_1 = 1$ and $s_1 = 0$, at session 100 (changes from session 1). **g** Transitions of selective neuronal responses of source 1-preferring neurons under control (middle), hypo- (left), and hyper-excitability (right) conditions. Red and blue lines represent the averaged evoked response of source 1-preferring neurons, when source 1 is ON or OFF, respectively ($n = 127, 514, 129$ electrodes from 7, 30, 6 independent experiments for diazepam, control, and bicuculline conditions, respectively). **h** Same as (**f**), but for empirical responses. Source data are provided as a Source Data file.

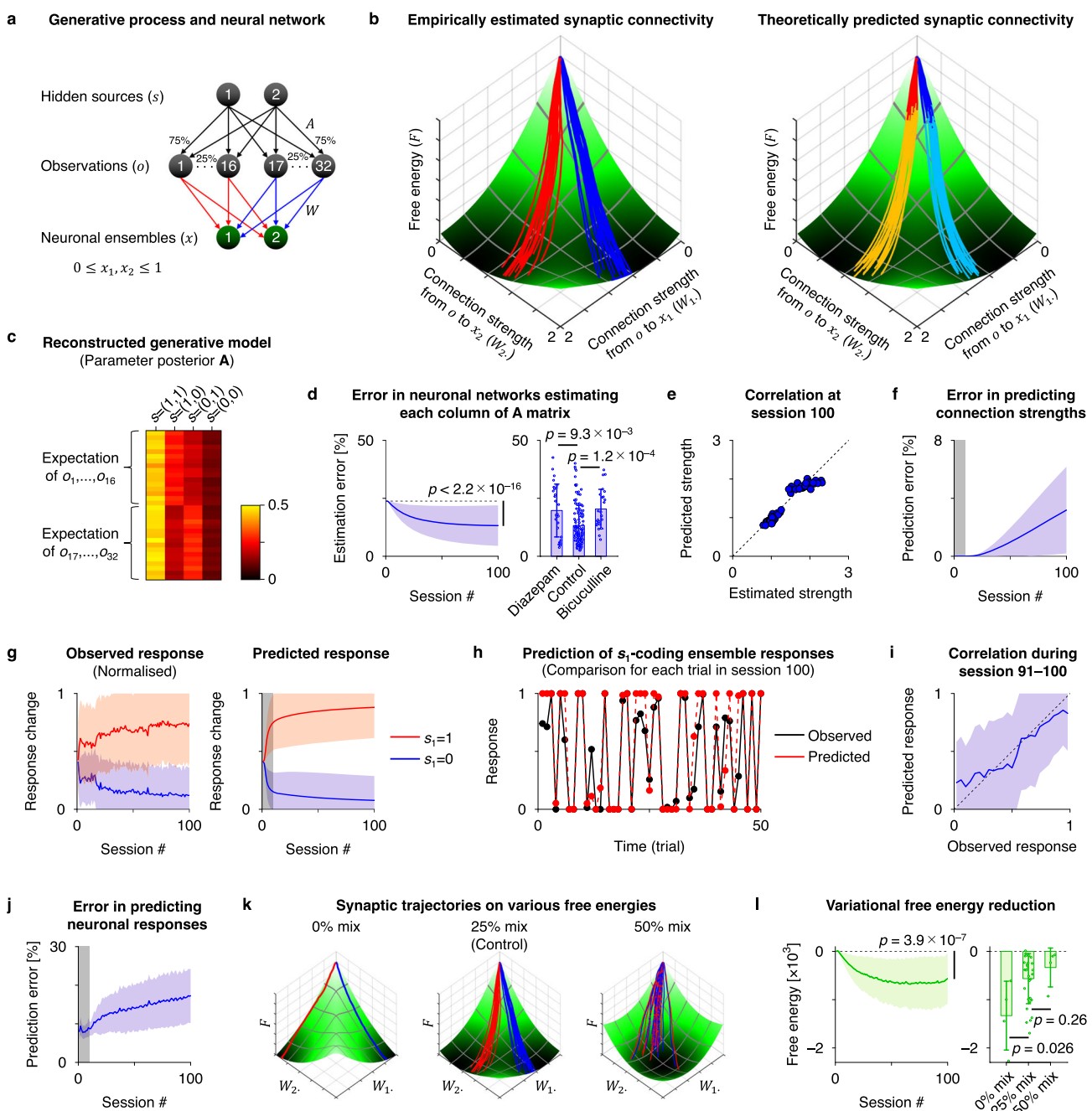

**Fig. 3 | Predictive validity of the free-energy principle. a** Schematic of the system architecture comprising the generative process and the in vitro *neuronal network* modelled as a canonical in silico *neural network*. Two neural ensembles receive 32 inputs generated from two sources. **b** Left: Trajectory of empirically estimated synaptic connectivity ($W$) depicted on the landscape of variational free energy ($F$). Red and blue lines show trajectories of red and blue connectivities in (**a**). The slope indicates a theoretically predicted free energy landscape. Darker green represents lower free energy. Whereas, synaptic strengths (i.e., effective synaptic connectivity or efficacy) are calculated using empirical data in sessions 1–100. Right: Predictions of synaptic plasticity during training. The initial conditions (i.e., parameters of a generative model) were identified using neuronal responses from the first 10 sessions. A brighter colour indicates the predicted synaptic trajectory in the absence of empirical response data. **c** Empirically estimated posterior belief (**A**) about parameter $A$. **d** Error in neuronal networks estimating each column of **A** matrix, defined as the squared error between empirical and ideal **A** matrices, divided by the squared amplitude of **A** ($n$ = 28, 120, 24 columns for diazepam, control, and biculline conditions, respectively). **e** Correlation between theoretically predicted

strengths and strengths estimated from data, at session 100. **f** Error in predicting synaptic strengths, defined as the squared error between estimated and predicted ($\widehat{W}_1, \widehat{W}_0$), divided by the squared Frobenius norm of ($\widehat{W}_1, \widehat{W}_0$) (see Methods for the definition). **g** Trajectory of observed (left) and predicted (right) neuronal responses of source 1-coding ensembles, during training. Red and blue lines indicate the responses when source 1 is ON and OFF, respectively. **h** Comparison of observed (black) and predicted (red) responses in session 100. **i** Correlation between observed and predicted responses during session 91–100. **j** Error in predicting neuronal responses, defined as the mean squared error: $err = E\left[|x_t - x_t^P|^2\right]/2$. **k** Synaptic trajectories on free energy landscape under 0, 25, and 50% mix conditions. **l** Trajectory of variational free energy. Changes from session 1 are plotted ($n$ = 4, 30, 4 independent experiments for 0, 25, and 50% mix conditions, respectively). In (**d**, **f**, **i**, **j**, **l**), data from $n$ = 30 independent experiments under the control condition were used. Lines and shaded areas (or error bars) in (**d**, **f**, **g**, **i**, **j**, **l**) represent mean values +/− standard deviations. Grey areas in (**f**, **g**, **j**) indicate the first 10 sessions, from which data were used. See Methods for further details. Source data are provided as a Source Data file.

corresponds to a conventional (model-based) connection strength estimation, where the $W$ of the canonical neural network model was optimised to fit the empirical data (see Methods). We then plotted the trajectory of the estimated synaptic strengths on the landscape of variational free energy $F$ (Fig. 3b, left). This landscape was characterised by the state prior (encoded by the firing threshold) estimated using empirical data from only the initial 10 sessions. According to the free-energy principle, synaptic plasticity occurs in a manner that descends on free energy gradients[1,2]. As predicted, we found that the trajectory of the empirically computed synaptic connectivity descended the free energy landscape (Fig. 3b, left; see also Supplementary Movie 1). This observation suggests that variational free energy minimisation is a plausible description of self-organisation or learning in neuronal networks.

Interestingly, the reverse engineering enables us to map empirically estimated synaptic strengths ($W$) to the posterior expectation (**A**) about parameter matrix $A$, to identify the generative model that the neuronal network employs (Table 2). The reconstructed posterior **A** precisely captured the characteristics of the true $A$ in the external milieu, such that source 1 has a greater contribution to $(o_1, \ldots, o_{16})$, while source 2 to $(o_{17}, \ldots, o_{32})$ (Fig. 3c). An error between empirical and ideal (Bayesian) posteriors significantly decreased with sessions (Fig. 3d, left). The error was larger in bicuculline- or diazepam-treated condition, owing to biased inference and subsequent learning in these neuronal networks (Fig. 3d, right). These results support the theoretical notion that synaptic strengths encode the posterior expectation of the parameter[1,2]. As predicted, synaptic plasticity following the free energy gradient entailed a recapitulation of the generative process within the neuronal network architecture.

The observation that the empirical synaptic trajectory pursues a gradient descent on variational free energy implies that one can predict the subsequent learning in the absence of empirical constraints. Once the initial values of synaptic strengths are identified, the subsequent learning process can in principle be predicted using the free-energy principle, under the canonical neural network (i.e., generative) model.

To test this hypothesis, we predicted the neuronal responses ($x$) and synaptic plasticity ($W$) in sessions 11–100 using the neural network cost function $L$ reconstructed based exclusively on the empirical responses in the initial 10 sessions (see the Methods section 'Data prediction using the free-energy principle' for details). As established above, this cost function is formally identical to variational free energy $F$ under a class of POMDP generative models[16,17]. Thus, evaluating the responses and plasticity that minimise this cost function $L$ ($\equiv F$)—in the absence of data—furnishes a prediction of neuronal responses and plasticity under the free-energy principle.

We found that the predicted changes in connectivity matched the changes in empirically estimated effective synaptic connectivity (Fig. 3b, right). Specifically, we observed a strong correlation between the synaptic strengths estimated using neuronal data and the strengths predicted in the absence of data (Fig. 3e). The prediction error was less than 4%, up to the final session (Fig. 3f; $n = 30$ independent experiments). These results indicate that, based on initial conditions, the free-energy principle can predict the self-organisation of neuronal networks.

In addition to the synaptic trajectory, we confirmed that a minimisation of free energy can predict the underlying changes in neuronal responses (Fig. 3g). The predictions based only on initial conditions were consistent with observed responses. Specifically, the predicted responses were consistent with the observed responses at each time step (Fig. 3h, i). Quantitatively, we could predict more than 80% of the neuronal responses in session 100, based only on data from sessions 1–10 (Fig. 3j). These results suggest that the free-energy principle can predict both changes in synaptic efficacy and the time evolution of neuronal responses based only on initial data. Note that this is a highly nontrivial prediction, because synaptic efficacy shows activity-dependent changes and neuronal responses depend upon synaptic efficacy.

Another interesting observation was that when we varied the free energy landscape by manipulating the mixing matrix $A$ in the stimulus generating process, empirical synaptic plasticity kept pursuing a gradient descent on the new variational free energy (Fig. 3k). This speaks to a generality of this physiological property. Here, we experimentally varied the mixing balance ($A$) of two sources between 0 and 50%, to train neuronal networks with the generated sensory stimuli ($o$), where 0% indicates an unmixed (i.e., easily separable) condition, while 50% indicates a uniformly mixed (i.e., inseparable) condition. Irrespective of various conditions (i.e., forms of generative process and prior beliefs), the reverse engineering could reconstruct generative models and predict subsequent self-organisations of neuronal networks (Supplementary Fig. 2; see also Supplementary Movie 1).

Finally, we observed that during the process of assimilating sensory information, neuronal networks significantly reduced their variational free energy (Fig. 3l). Here, variational free energy $F$ for each session was calculated empirically by substituting the observed neuronal responses into the cost function $L$. As expected, an easier task (i.e., 0% mix condition) entailed a faster (i.e., greater) reduction of variational free energy. These results provide explicit empirical evidence that neuronal networks self-organise to minimise variational free energy.

In summary, we found that the trajectory of the empirically estimated effective synaptic connectivity is consistent with a slow gradient descent on variational free energy. Furthermore, we demonstrated that the free-energy principle can quantitatively predict sensory learning in neuronal networks in terms of both neuronal responses and plasticity. These results suggest that the self-organisation of the neuronal networks—in response to structured sensory input—is consistent with Bayesian belief updating and the minimisation of variational free energy. This endorses the plausibility of variational free energy minimisation as a rule underlying the dynamics and self-organisation of neuronal networks.

## Discussion

The present work has addressed the predictive validity of the free-energy principle at the circuit level by delineating the functional specialisation and segregation in neuronal networks via free-energy minimisation. Identifying the characteristic functions of arbitrary neural networks is not straightforward. However, according to the complete class theorem[41–43], any system that minimises a cost function under uncertainty can be viewed as Bayesian inference. In light of this, we showed that any neural network—whose activity and plasticity minimise a common cost function—can be cast as performing (variational) Bayesian inference[16,17]. Crucially, the existence of this equivalence enables the identification of a natural map from neuronal activity to a unique generative model (i.e., hypothesis about the external milieu), under which a biological system operates. This step is essential to link empirical data—which report the 'internal' circuit dynamics (i.e., physiological phenomena)—to the representation of the 'external' dynamics (i.e., functions or computations) that the circuit dynamics imply, in terms of variational Bayesian inference. Using this technique, we fitted stimulus-evoked responses of in vitro networks—comprising the cortical cells of rat embryos—to a canonical neural network and reverse engineered an POMDP generative model, apt to explain the empirical data. In other words, we were able to explain empirical responses as inferring the causes of stimuli, under an implicit generative or world model.

Furthermore, reverse engineering a generative model from observed responses specifies a well-defined synaptic plasticity rule. Using this rule, we showed that the self-organisation of in vitro networks follows a gradient descent on variational free energy under the

(POMDP) generative model. In short, the virtues of reverse engineering are that: (i) when provided with empirical responses, it systematically identifies what hypothesis (i.e., generative model) the biological system employs to infer the external milieu. Moreover, (ii) it offers quantitative predictions about the subsequent self-organisation (i.e., learning) of the system that can be tested using data. This provides a useful tool for analysing and predicting electrophysiological and behavioural responses and elucidating the underlying computational and self-organisation principle.

Although numerous neural implementations of Bayesian inference have been proposed[44–46], these approaches generally derive update rules from Bayesian cost functions without establishing the precise relationship between these update rules and the neural activity and plasticity of canonical neural networks. The reverse engineering approach differs conceptually by asking what Bayesian scheme could account for any given neuronal dynamics or neural network. By identifying the implicit inversion scheme—and requisite generative model—one can then lend any given network an interpretability and explainability. In the current application of this approach, we first consider a biologically plausible cost function for neural networks that explains both neural activity and synaptic plasticity. We then identify a particular generative model under which variational free energy is equivalent to the neural network cost function. In this regard, reverse engineering offers an objective procedure for explaining neural networks in terms of Bayesian inference. Further, the synaptic plasticity rule is derived as the gradient descent on the cost function that is determined by the integral of neural dynamics. Crucially, learning through this plasticity rule can be read, formally, as Bayesian belief updating under an appropriate generative model. Conversely, naive Hebbian plasticity rules—with an ad hoc functional form—correspond to Bayesian belief updating under a suboptimal generative model with biased prior beliefs, which cannot solve simple blind source separation problems[16]. As predicted, in vitro neural networks above failed to perform blind source separation, with changed baseline excitability and implicit priors. In short, the free-energy principle is necessary to determine the optimal balance between Hebbian and homeostatic plasticity that enables blind source separation by in vitro networks.

Previous work has established that ensembles of neurons encode posterior expectations[47] and prediction errors[48]; however, other quantities in Bayesian inference—such as the state prior and parameter posterior—have yet to be fully investigated. The reverse engineering approach enables us to identify the structures, variables, and parameters of generative models from experimental data, which is essential for empirical applications of the free-energy principle. This is a notion referred to as computational phenotyping[49]; namely inferring the generative model—and in particular, the priors—that best explain empirical responses under ideal Bayesian assumptions. The reverse engineering naturally maps empirical (biological) quantities to quantities in variational Bayesian inference. Our empirical results suggest that neuronal responses encode the hidden state posterior (Fig. 2c), baseline excitability encodes the state prior (Fig. 2g), and synaptic efficacies encode the parameter posterior (Fig. 3c), as predicted theoretically (Table 2).

Having said this, because the free-energy principle can arguably describe any observed biological data by its construction[19], showing the existence of such a mapping alone is insufficient as an empirical validation. Conversely, one can examine the predictive validity, which is a more delicate problem, by asking whether the free-energy principle can predict subsequent self-organisation without reference to empirical data. Such a generalisability on previously unseen (test) data comprises an essential aspect for empirical applications of the free-energy principle.

We demonstrated that, equipped with the initial conditions (i.e., generative model and implicit prior beliefs of the network) characterised by the experimental data, variational free energy

minimisation can predict the subsequent self-organisation of in vitro neural networks, in terms of quantitative neuronal responses and plasticity. It further predicted their performance when spontaneously solving source separation problems, including their speed and accuracy. These results not only validate this application of the free-energy principle; they also speak to the neurophysiological plausibility of related theories of the brain[50,51] and spiking neural network models that perform Bayesian inference[44–46].

In essence, the free-energy principle constrains the relationship between neural activity and plasticity because both activity and plasticity follow a gradient descent on a common variational free energy, under ideal Bayesian assumptions. This property in turn enables precise characterisation of plausible self-organisation rules and quantitative prediction of subsequent neuronal activity and plasticity, under a canonical neural network (generative) model.

Our combined in vitro–in silico system showed that variation of the state prior (in silico model) is sufficient to reproduce the changes in neural excitability and inhibitions of sensory learning and inference observed in vitro. These results suggest that a neuronal networks' excitability is normally tuned so that the ensemble behaviour is close to that of a Bayes optimal encoder under biological constraints. This is reminiscent of previous experimental observation that suggests that the activity of sensory areas encodes prior beliefs[52].

These empirical data and complementary modelling results also explain the strong influence of prior beliefs on perception and causal inference—and the disruptive effects of drugs on perception in neuronal networks. Both synaptic plasticity and inference depend on convergent neuronal activity; therefore, aberrant inference will disrupt learning. Conversely, inference is not possible without the knowledge accumulated through experience (i.e., learning). Thus, inference is strongly linked to learning about contingencies that generate false inferences. Our findings demonstrate this association both mechanistically and mathematically, in terms of one simple rule that allows prior beliefs to underwrite inferences about hidden states.

Combining mathematical analyses with empirical observations revealed that baseline excitability is a circuit-level encoding of prior beliefs about hidden states. The notion that manipulating the state prior (encoded by neuronal excitability) disrupts inference and learning may explain the perceptual deficits produced by drugs that alter neuronal excitability, such as anxiolytics and psychedelics[53]. This may have profound implications for our understanding of how anxiolytics and psychedelics mediate their effects; namely, a direct effect on baseline activity can alter subsequent perceptual learning. Additionally, aberrant prior beliefs are a plausible cause of the hallucinations and delusions that constitute the positive symptoms of schizophrenia[54,55]. This suggests that, in principle, reverse engineering provides a formal avenue for estimating prior beliefs from empirical data—and for modelling the circuit mechanisms of psychiatric disorders (e.g., synaptopathy). Further, the reproduction of these phenomena in in vitro (and in vivo) networks furnishes the opportunity to elucidate the precise pharmacological, electrophysiological, and statistical mechanisms underlying Bayesian inference in the brain.

Importantly, although this paper focused on a comparison of in vitro data and theoretical prediction, the reverse engineering approach is applicable to characterising in vivo neuronal networks, in terms of their implicit generative model with prior beliefs. It can, in principle, be combined with electrophysiological, functional imaging, and behavioural data—and give predictions, if the learning process is continuously measured. Thus, the proposed approach for validating the free-energy principle can be applied to the neural activity data from any experiment that entails learning or self-organisation; irrespective of the species, brain region, task, or measurement technique. Even in the absence of learning, it can be applied, if one can make some theoretical predictions and compare them with experimental data. For accurate predictions, large-scale and continuous measurements of

activity data at the population level from pre-learning to post-learning stages would be a prerequisite. In future work, we hope to test, empirically, whether the free-energy principle can quantitatively predict the perception, learning, and behaviour of various biological systems.

The generic mechanisms for acquiring generative models can be used to construct a neuromorphic hardware for universal applications[56,57]. Back-propagation is central in many current deep learning methods, but biologically implausible. This has led to various biologically plausible alternatives; e.g., refs. [58–61], some of which appeal to predictive coding formulations of variational free energy minimisation. The equivalence between neural networks and variational Bayes could be useful to establish biologically plausible learning algorithms, because Hebbian learning rules derived under this scheme are local (energy-based) algorithms. This is because the contribution to variational free energy as an extensive quantity can be evaluated locally. Such a biomimetic artificial intelligence—that implements the self-organising mechanisms of neuronal networks—could offer an alternative to conventional learning algorithms such as back-propagation, and to have the high data, computational, and energy efficiency of biological computation[62,63]. This makes it promising for the next-generation of artificial intelligence. In addition, the creation of biomimetic artificial intelligence may further our understanding of the brain.

In summary, complementary in vitro neural network recordings and in silico modelling suggest that variational free energy minimisation is an apt explanation for the dynamics and self-organisation of neuronal networks that assimilate sensory data. The reverse engineering approach provides a powerful tool for the mechanistic investigation of inference and learning, enabling the identification of generative models and the application of the free-energy principle. The observed sensory learning was consistent with Bayesian belief updating and the minimisation of variational free energy. Thus, variational free energy minimisation could qualitatively predict neuronal responses and plasticity of in vitro neural networks. These results highlight the validity of the free-energy principle as a rule underlying the self-organisation and learning of neuronal networks.

## Methods

### Generative process

The experimental paradigm established in previous work[22] was employed. The blind source separation addressed in this work is an essential ability for biological organisms to identify hidden causes of sensory information, as considered in the cocktail party effect[27,28]. This deals with the separation of mixed sensory inputs into original hidden sources in the absence of supervision, which is a more complex problem than naive pattern separation tasks.

Two sequences of mutually independent hidden sources or states $s_t = (s_t^{(1)}, s_t^{(2)})^T$ generated 32 sensory stimuli $o_t = (o_t^{(1)}, \ldots, o_t^{(32)})^T$ through a stochastic mixture characterised by matrix $A$. Each source and observation took values of 1 (ON) or 0 (OFF) for each trial (or time) $t$. These stimuli were applied to in vitro neural networks as electrical pulses from 32 electrodes (Fig. 1a, left). In terms of the POMDP scheme[38,39], this corresponds to the likelihood mapping $A$ from two sources $s_t$ to 32 observations $o_t$ (Fig. 1a, right). The hidden sources $s_t$ were sampled from a categorical distribution $P(s_t^{(j)}) = \mathrm{Cat}(D^{(j)})$. The state priors varied between 0 and 1, in keeping with $D_1^{(j)} + D_0^{(j)} = 1$. The likelihood of $o_t^{(i)}$ is given in the form of a categorical distribution, $P(o_t^{(i)}|s_t, A) = \mathrm{Cat}(A^{(i)})$, each element of which represents $P(o_t^{(i)} = j|s_t^{(1)} = k, s_t^{(2)} = l, A) = A_{jkl}^{(i)}$. Half of the electrodes ($1 \le i \le 16$) conveyed the source 1 signal with a 75% probability or the source 2 signal with a 25% probability. Because each element of the $A$ matrix represents the conditional probability that $o_t$ occurs given $s_t = (s_t^{(1)}, s_t^{(2)})$, this

characteristic is expressed as $A_{1 \cdots}^{(i)} = (P(o_t^{(i)} = 1|s_t = (1,1)), P(o_t^{(i)} = 1|s_t = (1,0)), P(o_t^{(i)} = 1|s_t = (0,1)), P(o_t^{(i)} = 1|s_t = (0,0))) = (1, 0.75, 0.25, 0)$. The remaining electrodes ($17 \le i \le 32$) conveyed the source 1 or 2 signal with a 25% or 75% probability, respectively, $A_{1 \cdots}^{(i)} = (1, 0.25, 0.75, 0)$. The remaining elements of $A$ were given by $A_{0 \cdots}^{(i)} = 1 - A_{1 \cdots}^{(i)}$. The prior distribution of $A$ was given by the Dirichlet distribution $P(A^{(i)}) = \mathrm{Dir}(a^{(i)})$ with sufficient statistics $a$. Hence, the generative model was given as follows:

$$P(o_{1:t}, s_{1:t}, A) = P(A) \prod_{\tau=1}^{t} P(s_\tau) P(o_\tau|s_\tau, A) \tag{1}$$

Here, $o_{1:t} := \{o_1, \ldots, o_t\}$ represents a sequence of observations, $P(A) = \prod_{i=1}^{32} P(A^{(i)})$, $P(s_\tau) = P(s_\tau^{(1)}) P(s_\tau^{(2)})$, and $P(o_\tau|s_\tau, A) = \prod_{i=1}^{32} P(o_\tau^{(i)}|s_\tau, A)$ are prior distributions and likelihood that factorise[16].

### Variational Bayesian inference

We considered a Bayesian observer under the generative model in the form of the above POMPD and implemented variational message passing to derive the Bayes optimal encoding of hidden sources or states[38,39]. Under the mean-field approximation, the posterior beliefs about states and parameters were provided as follows:

$$Q(s_{1:t}, A) = Q(A) \prod_{\tau=1}^{t} Q(s_\tau) \tag{2}$$

Here, the posterior distributions of $s_\tau$ and $A$ are given by categorical $Q(s_\tau) = \mathrm{Cat}(\mathbf{s}_t)$ and Dirichlet $Q(A) = \mathrm{Dir}(\mathbf{a})$ distributions, respectively. The bold case variables (e.g., $\mathbf{s}_t$) denote the posterior beliefs about the corresponding italic case variables (e.g., $s_t$), and $\mathbf{a}$ indicates the Dirichlet concentration parameter. Due to the factorial nature of the states, $\mathbf{s}_t$ and $\mathbf{a}$ are the outer products of submatrices (i.e., tensors): see ref. 16 for details.

Variational free energy—or equivalently, the negative of evidence lower bound (ELBO)[3]—is defined as an upper bound of sensory surprise $F(o_{1:t}, Q(s_{1:t}, A)) := E_{Q(s_{1:t}, A)}[-\ln P(o_{1:t}, s_{1:t}, A) + \ln Q(s_{1:t}, A)]$. Given the above-defined generative model and posterior beliefs, ensuing variational free energy of this system is given by:

$$F = \sum_{\tau=1}^{t} \mathbf{s}_\tau \cdot (\ln \mathbf{s}_\tau - \ln \mathbf{A} \cdot o_\tau - \ln D) + \mathcal{O}(\ln t) \tag{3}$$

up to an $\mathcal{O}(\ln t)$ term. This $\mathcal{O}(\ln t)$ corresponds to the parameter complexity expressed using the Kullback–Leibler divergence $\mathcal{D}_{KL}[Q(A)||P(A)] = \sum_{i=1}^{32} \{(\mathbf{a}^{(i)} - a^{(i)}) \cdot \ln \mathbf{A}^{(i)} - \ln \mathcal{B}(\mathbf{a}^{(i)})\}$ and is negligible when $t$ is sufficiently large. Note that $\cdot$ expresses the inner product operator, $\ln \mathbf{A}^{(i)}$ indicates the posterior expectation of $\ln A^{(i)}$, and $\mathcal{B}(\bullet)$ is the beta function. Inference and learning entail updating posterior expectations about hidden states and parameters, respectively, to minimise variational free energy. Solving the fixed point $\partial F/\partial \mathbf{s}_t = 0$ and $\partial F/\partial \mathbf{a} = O$ yields the following analytic expression:

$$\mathbf{s}_t = \sigma(\ln \mathbf{A} \cdot o_t + \ln D) \tag{4}$$

$$\mathbf{a} = a + \sum_{\tau=1}^{t} o_\tau \otimes \mathbf{s}_\tau \tag{5}$$

where $\sigma(\bullet)$ is a softmax function, which corresponds to the sigmoid activation function, and $\otimes$ expresses the outer product operator. From Eq. (5), the parameter posterior is given as $\ln \mathbf{A} = \psi(\mathbf{a}) - \psi(\mathbf{a}_{1\bullet} + \mathbf{a}_{0\bullet})$ using the digamma function $\psi(\bullet) = \Gamma'(\bullet)/\Gamma(\bullet)$. As Eqs.

(2)–(5) adopted a simplified notation, please refer to ref. 16 for the detailed derivation taking into account the factorial nature of the states.

## Canonical neural networks

The complete class theorem[41–43] suggests that any neural network whose internal states minimise a common cost function can be read as performing Bayesian inference. However, the implicit Bayesian model that corresponds to any given cost function is a more complicated issue. Thus, we reverse engineered cost functions for canonical neural networks to identify the corresponding generative model[16,17].

The neural response $x_t = (x_{t1}, x_{t2})^T$ at time $t$ upon receiving sensory inputs $o_t$ is modelled as the canonical neural network, which is expressed as the following ordinary differential equation:

$$\dot{x}_t \propto -\mathrm{sig}^{-1}(x_t) + W o_t + h \qquad (6)$$

where $\mathrm{sig}^{-1}(x_t)$ indicates the leak current characterised by the inverse of sigmoid function (or equivalently, logit function), $W$ denotes a $2 \times 32$ matrix of synaptic strengths, $W o_t$ is the synaptic input, and $h$ is a vector of the adaptive firing thresholds. We considered that $W := W_1 - W_0$ is the sum of excitatory ($W_1$) and inhibitory ($W_0$) synaptic strengths. The firing threshold is expressed as $h := h_1 - h_0$ using $h_1$ and $h_0$ that are functions of $W_1$ and $W_0$. This model derives from realistic neuron models[4–6] through approximations[17].

Without loss of generality, Eq. (6) can be derived as the gradient descent on a cost function $L$. Following previous work[16,17], this cost function can be identified by taking the integral of the right-hand side of Eq. (6) with respect to $x_t$ (referred to as reverse engineering):

$$L = \sum_{\tau=1}^{t} \left( \frac{x_\tau}{\overline{x_\tau}} \right)^T \left\{ \ln\left( \frac{x_\tau}{\overline{x_\tau}} \right) - \binom{W_1}{W_0} o_\tau - \binom{h_1}{h_0} \right\} + \mathcal{C} \qquad (7)$$

up to a negligible $\mathcal{C}$ term. The overline variable indicates one minus the variable, $\overline{x_t} := \vec{1} - x_t$, where $\vec{1} := (1, \ldots, 1)^T$ is a vector of ones. Equation (7) ensures that the gradient descent on $L$ with respect to $x_t$, $\dot{x}_t \propto -\partial L/\partial x_t$, provides Eq. (6). The $\mathcal{C}$ term is a function of $W_1$ and $W_0$, $\mathcal{C} = \mathcal{C}(W_1, W_0)$, and usually considered to be smaller than order $t$, $\mathcal{C} = o(t)$[16].

The firing thresholds can be decomposed as $h_1 = \ln \widehat{\overline{W_1}} \vec{1} + \phi_1$ and $h_0 = \ln \widehat{\overline{W_0}} \vec{1} + \phi_0$, respectively, where $\phi_1$ and $\phi_0$ are the threshold factors, $\widehat{W_1} := \mathrm{sig}(W_1)$ is the sigmoid function of $W_1$ in the elementwise sense, and $\widehat{\overline{W_1}}$ indicates one minus $\widehat{W_1}$ in the elementwise sense. Subsequently, Eq. (7) can be transformed as follows:

$$L = \sum_{\tau=1}^{t} \left( \frac{x_\tau}{\overline{x_\tau}} \right)^T \left\{ \ln\left( \frac{x_\tau}{\overline{x_\tau}} \right) - \ln\begin{pmatrix} \widehat{W_1} & \widehat{\overline{W_1}} \\ \widehat{W_0} & \widehat{\overline{W_0}} \end{pmatrix} \begin{pmatrix} o_\tau \\ \overline{o_\tau} \end{pmatrix} - \begin{pmatrix} \phi_1 \\ \phi_0 \end{pmatrix} \right\} + \mathcal{C} \qquad (8)$$

We showed that this cost function $L$ can be cast as variational free energy $F$ under a class of POMPD generative models[16,17]. Equation (8) is asymptotically equivalent to variational free energy (Eq. (3)) under the generative model defined in Eq. (1), up to negligible $\mathcal{O}(\ln t)$ and $\mathcal{C}$ terms. One-to-one correspondences between components of $L$ and $F$ can be observed. Specifically, the neural response $x_t$ encodes the state posterior $\mathbf{s}_t$, $\left( \frac{x_\tau}{\overline{x_\tau}} \right) = \mathbf{s}_\tau$; synaptic strengths $W$ encode the parameter posterior $\mathbf{A}$, $\ln\begin{pmatrix} \widehat{W_1} & \widehat{\overline{W_1}} \\ \widehat{W_0} & \widehat{\overline{W_0}} \end{pmatrix} = \ln \mathbf{A}$; and the threshold factor $\phi$ encodes the state prior $D$, $\phi = \begin{pmatrix} \phi_1 \\ \phi_0 \end{pmatrix} = \ln D$, as summarised in Table 2. Hence, the neural network cost function is asymptotically equivalent to variational free energy for sufficiently large $t$. Further details, including the correspondence between the $\mathcal{C}$ term in Eq. (8) and the parameter complexity ($\mathcal{O}(\ln t)$ term in Eq. (3)), are described in previous work[16].

The virtue of this equivalence is that it links quantities in the neural network with those in the variational Bayes formation. Moreover, this suggests that a physiologically plausible synaptic plasticity (derived from $L$) enables the network to learn the parameter posterior in a self-organising or unsupervised manner[16,17]. Further, reverse-engineering can naturally derive variational Bayesian inference—under a particular mean-field approximation defined in Eq. 2—from a canonical neural network architecture. This representation of posterior beliefs is essential for the networks to encode rapidly changing hidden states ($s_\tau$) and slow parameters ($A$) with neural activity ($x$) and synaptic strengths ($W$), respectively. In this setting, a mean field approximation implements a kind of adiabatic approximation, in which the separation of timescales between fast neuronal responses and slow learning is leveraged to increase the efficiency of inference. Please see ref. 16. for further discussion.

## Simulations

In Fig. 2, simulations continued over $T = 25600$ time steps and used the empirical stimuli applied to in vitro neural networks. Synaptic strengths were initialised as values close to 0. Here, $D_1 = 0.5$ (Fig. 2e, centre) matched the true process that generates sensory stimuli. Either the upregulation (right, $D_1 = 0.8$) or downregulation (left, $D_1 = 0.2$) of the state prior disrupted inference and ensuing learning.

## Cell culture

The dataset used for this work comprised data obtained from newly conducted experiments, and those originally used in the previous work[22]. All animal experiments were performed with the approval of the animal experiment ethics committee at the University of Tokyo (approval number C-12-02, KA-14-2) and according to the University of Tokyo guidelines for the care and use of laboratory animals. The procedure for preparing dissociated cultures of cortical neurons followed the procedures described in previous work[22]. Pregnant Wistar rats (Charles River Laboratories, Yokohama, Japan) were anaesthetised with isoflurane and immediately sacrificed. The cerebral cortex was removed from 19-day-old embryos (E19) and dissociated into single cells by treatment with 2.5% trypsin (Life Technologies, Carlsbad, CA, USA) at 37 °C for 20 min, followed by mechanical pipetting. Half a million dissociated cortical cells (a mixture of neurons and glial cells) were seeded on the centre of MEA dishes, where the surface of MEA was previously coated with polyethyleneimine (Sigma–Aldrich, St. Louis, MO, USA) overnight. These cells were cultured in the $CO_2$ incubator. Culture medium comprised Neurobasal Medium (Life Technologies) containing 2% B27 Supplement (Life Technologies), 2 mM GlutaMAX (Life Technologies), and 5–40 U/mL penicillin/streptomycin (Life Technologies). Half of the culture medium was changed once every second or third day. These cultures were recorded during the age of 18–83 days in vitro. During this stage, the spontaneous firing patterns of the neurons had reached a developmentally stable period[64,65].

In this work, 21 independent cell cultures were used for the control condition to conduct 30 independent experiments, 6 were treated with bicuculline, 7 with diazepam, 9 with APV, 4 were trained under the 0% mix condition, and 4 under the 50% mix condition. Out of these samples, 7 in the control condition, 6 treated with bicuculline, and 7 with diazepam were obtained from newly conducted experiments, where their response intensities were $3.0 \pm 1.1$, $3.7 \pm 1.9$, and $2.3 \pm 0.86$ spike/trial, respectively (mean ± standard deviation). Other cultures were originally recorded for previous work[22]. The cell-culturing and experimental conditions in the previous work were essentially the same as those recorded for the present work. Note that the same cultures were used more than once for experiments with other stimulation pattern conditions, after at least one day interval.

This is justified because the different stimulation patterns were independent of each other, and thus, learning history with other stimulation patterns did not affect the subsequent experiments[22].

## Pharmacological treatment
The excitability of cultured neurons was pharmacologically controlled. To block $GABA_A$-receptor activity, bicuculline, a $GABA_A$-receptor antagonist (Sigma–Aldrich, St. Louis, MO, USA) was used. Bicuculline was adjusted to 10 mM using phosphate-buffered saline (PBS), and 10 μL was added to the culture medium to a final concentration of 50 μM. To upregulate $GABA_A$-receptor activity, diazepam, a benzodiazepine receptor agonist (Sigma–Aldrich) was used. Diazepam was adjusted to 100 μM using N,N-dimethylformamide (DMF), and 20 μL was added to the culture medium to a final concentration of 1 μM. After adding the solution to the medium, cultured neurons were placed in a $CO_2$ incubator for 30 min, and stable activity of the neurons was confirmed before recording.

## Electrophysiological experiments
Electrophysiological experiments were conducted using an MEA system (NF Corporation, Yokohama, Japan). This enabled extracellular recordings of evoked spikes from multiple sites immediately after electrical stimulation[20,21]. An MEA dish comprises 8×8 microelectrodes (50-μm square each) embedded on its centre, deployed on a grid with 250-μm microelectrodes separation. Recordings were conducted with a 25-kHz sampling frequency and band-pass filter of 100–2000 Hz. The data acquisition was conducted using LabVIEW version 2011. The spike sorting analysis suggested that an electrode was expected to record the activities from up to four neurons. Three-phasic extracellular potentials, as described in previous work[20,21], were recorded from the majority of the electrodes.

The 32 stimulation electrodes were randomly distributed over 8×8 MEAs in advance and fixed over training. A biphasic pulse with a 1-V amplitude and 0.2-ms duration, which efficiently induces activity-dependent synaptic plasticity[22], was used as a sensory stimulus. A session of training comprised a 256-time-step sequence of stimuli with 1-s intervals, followed by a 244-s resting period. We repeated this training for 100 sessions (approximately 14 h in total). All recordings and stimulation were conducted in a $CO_2$ incubator.

## Data preprocessing
For spike detection, the acquired signals were passed through a digital band-pass filter of 500–2000 Hz after the removal of the saturated ranges and noises that were caused by electric potential variations associated with the switch from the stimulation circuit to the recording circuit. Subsequently, waveform valleys that fell below 4 times the standard deviation of the signal sequence of each electrode were detected as spikes. Note that for data obtained in the previous work[22], waveform valleys that fell below 5 times the standard deviation were detected as spikes because of the difference in the noise level.

Irrespective of the presence or absence of bicuculline or diazepam, the peak of evoked response usually fell at 10–20 ms after each stimulus. Accordingly, we defined the intensity of the evoked response to the stimulus by the number of spikes generated until 10–30 ms after each stimulus. We referred to the evoked response at electrode $i$ as $r_{ti}$ (spike/trial), using discrete time step (or trial) $t$. Only electrodes at which the all-session average of $r_{ti}$ was larger than 1 spike/trial were used for subsequent analyses.

The conditional expectation of evoked response $r_{ti}$—when a certain source state $(s_1,s_2) = (1,1),(1,0),(0,1),(0,0)$ is provided—is given as $E[r_{ti}|s_1,s_2] := E[r_{ti}|s_t = (s_1,s_2), 1 \leq t \leq 256]$ (spike/trial). This $E[r_{ti}|s_1,s_2]$ was computed for each session. Recorded neurons were categorised into three groups based on their preference to sources. We referred to a neuron (or electrode) as source 1-preferring when the all-session average of $E[r_{ti}|1,0] - E[r_{ti}|0,1]$ was larger than 0.5 spike/trial, as

source 2-preferring when the all-session average of $E[r_{ti}|1,0] - E[r_{ti}|0,1]$ was smaller than −0.5 spike/trial, or no preference when otherwise. Note that the number of source 1-preferring, source 2-preferring, and no preference electrodes in each sample are $17.1 \pm 7.0$, $15.0 \pm 7.0$, and $11.5 \pm 6.7$, respectively ($n = 30$ samples under the control condition). Sources 1- and 2-preferring ensembles were quantitatively similar because the total contribution from sources 1 and 2 to stimuli $o_t$ was designed to be equivalent, owing to the symmetric structure of the $A$ matrix. Under this setting, this similarity was conserved, irrespective of the details of the $A$.

Our hypothesis[23] was that the stimulus ($o$) obligatorily excites a subset of neurons in the network, while repeated exposure makes other neurons with appropriate connectivity learn that the patterns of responses are caused by ON or OFF of hidden sources ($s$). Thus, the recorded neuronal responses comprise the activity of neurons directly receiving the input and that of neurons encoding the sources. To identify functionally specialised neurons, we modelled recorded activity as a mixture of the response directly triggered by the stimulus and functionally specialised response to the sources. Most directly triggered responses occur within 10 ms of stimulation, and their number is largely invariant over time, while their latency varies in the range of a few hundred microseconds[40]. Conversely, functionally specialised responses emerge during training, and the majority occur 10–30 ms after stimulation. Thus, analysing the deviation of the number of spikes in this period enables the decomposition of the responses into stimulus- and source-specific components.

The empirical responses were represented as the averaged responses in each group. For subsequent analyses, we defined $x_{t1}$ as the ensemble average over source 1-preferring neurons and $x_{t2}$ as that over source 2-preferring neurons in each culture. For analytical tractability, we normalised the recorded neural response to ensure that it was within the range of $0 \leq x_{t1}, x_{t2} \leq 1$, after subtracting the offset and trend.

## Statistical tests
The two-sided Wilcoxon signed-rank test was used for paired comparisons. The two-sided Mann–Whitney $U$ test was used for unpaired comparisons.

## Reverse engineering of generative models
In this section, we elaborate the procedure for estimating the threshold factor ($\phi$) and effective synaptic connectivity ($W$) from empirical data, to characterise the landscape of the neural network cost function $L$ ($\equiv F$) and further derive the generative model that the biological system employs.

Assuming that the change in threshold factor was sufficiently slow relative to a short experimental period, the threshold factor $\phi$ was estimated based on the mean response intensity of empirical data. Following the treatment established in previous work[16,17], the constants are estimated for each culture using the empirical data as follows:

$$\phi = \begin{pmatrix} \phi_1 \\ \phi_0 \end{pmatrix} = \ln\left( \frac{\langle x_t \rangle}{\langle \overline{x_t} \rangle} \right) \tag{9}$$

where $\langle \cdot \rangle := \frac{1}{t}\sum_{\tau=1}^{t} \cdot$ indicates the average over time. Equation (9) was computed using data in the initial 10 sessions. Subsequently, the state prior $D$ was reconstructed from the relationship $\ln D = \phi$ (Table 2). This $D$ expresses the implicit perceptual bias of an in vitro network about hidden sources.

Synaptic plasticity rules conjugate to Eq. (6) are derived as the gradient descent on $L$[16,17], which are asymptotically given as $\dot{W}_1 \propto -\frac{1}{t}\frac{\partial L}{\partial W_1} = \langle x_t o_t^T \rangle - \langle x_t \vec{1}^T \rangle \odot \hat{W}_1$ and $\dot{W}_0 \propto -\frac{1}{t}\frac{\partial L}{\partial W_0} = \langle \overline{x_t} o_t^T \rangle - \langle \overline{x_t} \vec{1}^T \rangle \odot \hat{W}_0$ in the limit of a large $t$, where $\odot$ denotes the elementwise product (a.k.a., the Hadamard product). These rules comprise Hebbian plasticity accompanied with an activity-dependent homeostatic term,

endorsing the biological plausibility of this class of cost functions. Solving the fixed point of these equations provides the following synaptic strengths:

$$\begin{cases} W_1 = \text{sig}^{-1}\left(\langle x_t o_t^{\mathsf{T}} \rangle \oslash \langle x_t \vec{1}^{\mathsf{T}} \rangle\right) \\ W_0 = \text{sig}^{-1}\left(\langle \overline{x_t} o_t^{\mathsf{T}} \rangle \oslash \langle \overline{x_t} \vec{1}^{\mathsf{T}} \rangle\right) \end{cases} \tag{10}$$

where $\oslash$ denotes the elementwise division operator. In this work, we refer to Eq. (10) as the empirically estimated effective synaptic connectivity, where $W = W_1 - W_0$. This was estimated for each session, using empirical neuronal response data $x_t$. These synaptic strengths encode the posterior belief about the mixing matrix $A$ (Table 2). Further details are provided in previous works[16,17].

These empirically estimated parameters are sufficient to characterise the generative model that an in vitro neural network employs. Owing to the equivalence, the empirical variational free energy $F$ for the in vitro network was computed by substituting empirical neuronal responses $x$ and empirically estimated parameters $W$ (Eq. (10)) and $\phi$ (Eq. (9)) into the neural network cost function $L$ (Eq. (8)): see Fig. 3l for its trajectory.

### Data prediction using the free-energy principle

The virtues of the free-energy principle are that it offers the quantitative prediction of transitions (i.e., plasticity) of the neural responses and synaptic strengths in future, in the absence of empirical response data. We denote the predicted neural responses and synaptic strengths as $x_t^P$ and $W^P$, respectively, to distinguish them from the observed neural responses $x_t$ and empirically estimated synaptic strengths $W$ defined above.

The predicted neural response is given as the fixed-point solution of Eq. (6):

$$x_t^P = \text{sig}\left(W^P o_t + h^P\right) \tag{11}$$

where $h^P = \overline{\ln \widehat{W}_1^P \vec{1}} - \overline{\ln \widehat{W}_0^P \vec{1}} + \phi_1 - \phi_0$ denotes the adaptive firing threshold. Empirical $\phi$ (Eq. (9)) estimated from data in the initial 10 sessions was used to characterise $h^P$. Here, predicted synaptic strength matrix $W^P$ was used instead of the empirically estimated $W$. The predicted synaptic strengths are given as follows:

$$\begin{cases} W_1^P = \text{sig}^{-1}\left(\langle x_t^P o_t^{\mathsf{T}} \rangle \oslash \langle x_t^P \vec{1}^{\mathsf{T}} \rangle\right) \\ W_0^P = \text{sig}^{-1}\left(\langle \overline{x_t^P} o_t^{\mathsf{T}} \rangle \oslash \langle \overline{x_t^P} \vec{1}^{\mathsf{T}} \rangle\right) \end{cases} \tag{12}$$

where $W^P := W_1^P - W_0^P$. Here, the predicted neural responses $x_t^P$ were employed to compute the outer products. The initial value of $W^P$ was computed using empirical response data in the first 10 sessions. By computing Eqs. (11) and (12), one can predict the subsequent self-organisation of neuronal networks in sessions 11–100, without reference to the observed neuronal responses.

We note that the reverse engineering approach provides three novel aspects compared to earlier work[22,23]. First, previous work assumed the form of the generative model and did not examine whether all elements of the generative model corresponded to biological entities at the circuit level. In the present work, we objectively reverse-engineered the generative model from empirical data and showed a one-to-one mapping between all the elements of the generative model and neural network entities. Second, previous work did not examine the impact of changing prior beliefs on Bayesian inference performed by in vitro neural networks. The present work analysed how Bayesian inference and free energy reduction changed when the prior belief and external environment were artificially manipulated and showed that the results were consistent with theoretical predictions. This work validated the predicted relationship between baseline

excitability and prior beliefs about hidden states. Third, previous work did not investigate whether the free-energy principle can quantitatively predict the learning process of biological neural networks based exclusively on initial empirical data. This was demonstrated in the current work.

### Reporting summary

Further information on research design is available in the Nature Portfolio Reporting Summary linked to this article.

## Data availability

The neuronal response data are available at GitHub https://github.com/takuyaisomura/reverse_engineering. Source data are provided with this paper.

## Code availability

The simulations and analyses were conducted using MATLAB version R2020a. The scripts are available at GitHub https://github.com/takuyaisomura/reverse_engineering[66]. The scripts are covered under the GNU General Public License v3.0.

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

## Acknowledgements

We are grateful to Masafumi Oizumi, Asaki Kataoka, Daiki Sekizawa, and other members of the Oizumi laboratory for discussions. T.I. is supported by the Japan Society for the Promotion of Science (JSPS) KAKENHI Grant Numbers JP23H04973 and JP23H03465 and the Japan Science and Technology Agency (JST) CREST Grant Number JPMJCR22P1. K.J.F. is supported by funding for the Wellcome Centre for Human Neuroimaging (Ref: 205103/Z/16/Z), a Canada-UK Artificial Intelligence Initiative (Ref: ES/T01279X/1) and the European Union's Horizon 2020 Framework Programme for Research and Innovation under the Specific Grant Agreement No. 945539 (Human Brain Project SGA3). The funders had no role in study design, data collection and analysis, decision to publish, or preparation of the manuscript.

## Author contributions

Conceptualisation, T.I., K.K., Y.J. and K.J.F.; Designing and performing experiments and data analyses, T.I.; Writing—original draft, T.I. and K.J.F.; Writing—review & editing, T.I., K.K., Y.J. and K.J.F. All authors made substantial contributions to the conception, design and writing of the article. All authors have read and agreed to the published version of the manuscript.

## Competing interests

The authors declare no competing interests.
