## [Peer Review File · Nature Communications]

Experimental validation of the free-energy principle with in vitro neural networksREVIEWER COMMENTS

Reviewer #1 (Remarks to the Author):

The ability to train a living neuronal network to do blind source separation is a noteworthy result of this paper, and I believe the experimental work is sound and supports that claim, with caveats below. I am not sure if this is a new result or not, as other similar papers have been published, including by these authors (e.g., Isomura et al. (2015)). I believe the goal of this paper, however, was to help establish the Free Energy Principle as a useful tool with which to predict eventual learning by neuronal networks given initial learning and activity results. I am not well qualified to judge whether this was accomplished successfully, as Bayesian statistics are not my field of expertise. It would be helpful to define your technical terms carefully for those in other fields who may not be familiar with variational Bayes or FEP. The information in Table 1 is a good start. That said, I will offer some suggestions for improvements of this paper, based on the parts I can follow.

Neurobiologists would appreciate any new tools to help understand what is going on in living neuronal networks. Has this set of experiments told us anything new about how in vitro networks (and presumably, in vivo ones) learn? If so, please make it more clear what those advances are. The evidence that applying GABAergic blockers and agonists reduces learning should be bolstered with more controls, and especially, to eliminate the possibility that the network has just been disabled unnaturally by these treatments.

In several places (e.g., line 427), reference is made to single-cell or synaptic changes, while the experimental methods described do not address any individual synapses, let alone single cells. I suggest you sort spikes and follow the excitability of sorted units (individual neurons) recorded and stimulated. To get the ground truth of whether a synapse is strengthened or weakened, one must carry out intracellular recordings or perhaps calcium or voltage imaging at the single-cell level. I suggest you refer to changes you have observed in network excitability as a network phenomenon, and steer clear of making any claims about individual cells or synapses unless you do those types of single-cell experiments. I believe your data clearly show you can alter responses of certain (unspecified) circuits within the network, to allow blind source separation. Presumably, those are due to changes in synaptic strengths, but you have no data to address what is going on at individual synapses, and many alternative possible mechanisms exist for changing the response properties of a neuronal circuit. It would be helpful if you were to differentiate (by adding several synaptic blockers, e.g.) between post-synaptic responses to stimulation and directly evoked action potentials in the stimulated cells. Contrary to what you claimed on line 699, even the direct responses are prone to vary over time.

In lines 252-254, it is mistakenly said that GABAergic inputs are upregulated by a GABA antagonist and downregulated by a GABA agonist. This is backwards from what these drugs do. I think you meant to say "network excitability" rather than GABAergic inputs.

You have overstated (e.g., lines 230, 271, 280) the effects of these drugs (and APV), when your data show they reduce the learning effects, while some learning persists. It would be good if you could separate the performance of the network, while under the influence of blockers and agonists, from their effects on learning by only testing their effects on learning after the drugs have been washed out. Otherwise, you have merely shown that you can disrupt network activity, not learning, which almost any chemical can do.

Reviewer #2 (Remarks to the Author):

The paper describes a thorough experimental validation of the free-energy principle for quantitative prediction of neuronal network responses of in-vitro neuronal cultures and in-silico neural network simulations. Pharmacology is used to show that changes in synaptic connectivity affects variational free energy.

The work builds on the authors former publications (ref 20-23), and here significantly extends it, by showing an experimental validation of the quantitative prediction of self-organization and trajectory of plasticity of in-vitro neuronal networks as measured with microelectrode arrays. Especially interesting and novel is the pharmacological manipulations demonstrating that the effects of compounds on excitability are very well aligned between the in-vitro model and the in-silico simulations using this method.

The method is well described and thorough, providing sufficient information to reproduce it by other groups. I recommend the work to be published with minor corrections, as listed below:

- L212: provide more details on how these stimuli were “carefully constructed”
- L219: for clarity, specify explicitly that each of the 100 session was an identical repetition of the “random” sequence within each 256s long session.
- L222: please provide some justification on the selected reponses window of 10-30ms (or reference older work where this was justified).
- L225: Additional data showing how stable this “classification” into the tow groups was would be very interesting. Was this classification dependent on the carefully constructed o? Did it change over time? Any spatial correlation to o? Was it for all conditions roughly an equal number of electrodes in each group, and it not, was this somehow accounted for?

- L223: Correct to assume no spike sorting was performed, and “neurons” here actually correspond to “electrode”, with spikes from potentially multiple neurons mixed together. If so, please clarify in the text. Same also e.g. on L233, and other places in the MS.
- Fig2c: y-axis unit is likely not “spike/trial”, but unit-less, as it is showing the “change”. Correct? Please correct, also in all other similar figures.
- Fig2a: correct that the white raster plot for $s=(0,0)$ is showing spontaneous activity? Why is there not even a single spike in this section?
- Method Cell Culture: was there a primary or secondary coating used? Please specify.

Reviewer #3 (Remarks to the Author):

In earlier work, the authors show the connection between cost minimization in neural networks on the one hand and minimizing variational free energy on the other hand. This relationship enables the interpretation of learning in neural networks as variational inference.

In the current paper, the authors extend this work to show that variational free energy minimisation can quantitatively *predict* the self-organisation of neuronal networks in terms of their responses and plasticity.

Major comments:

1. It is probably fair to say that predicting the plasticity and responses in the biological neural networks can be achieved by simulating the corresponding artificial neural networks, by implementing (6) and Hebbian dynamics for the weight updates. Therefore, it is not the case that the variational inference approach is the only way to predict the synaptic weights and responses. However, it is indeed a nice interpretation of the latter in terms of inference. But I would argue that neural computations have been interpreted as Bayesian inference before by numerous researchers. What is precisely the novelty in this paper? Is the fact that one can do numerical predictions and interpret the variables in the neural net in terms of random variables that are being inferred? Of course, the inference problem at hand is rather simple, therefore, there might not be much benefit from viewing the computations as inference here.
2. I think it would be interesting to speculate about contexts where the inference viewpoint is essential to understand the phenomenon. This could be in the context of cognitive psychology or psychiatry, where changing priors leads to different behaviors (potentially erratic ones). Could the authors perhaps extend the inference problem at hand to such scenario? For instance, binocular rivalry, where there also seem to be two "sources" or two competing explanations. Having such illustration/example may better underline the potential of the formalism introduced in the paper, since the considered inference problem is a simple and abstract, and a too distant from realistic inference problems.

3. The authors rely on variational inference. Is it possible that other inference mechanisms would to similar results and interpretations, e.g., Monte Carlo or belief propagation? I don't understand why the mean field approach is essential. Or is it essential? Could also more complex mean field methods (structured mean field) be mapped to neural computations?

4. The paper only deals with in vitro biological neural networks. Can the conclusions be generalized to in vivo neural networks? Can similar experiments be conducted in vivo? What would be the challenges? I recommend the authors to reflect on this topic in the Discussion section.

Minor comments:

line 84: "as neuronal networks and reserve the term neural network for an in silico model" -> I would suggest the terminology "biological neural networks (BNN)" and "artificial neural networks (ANN)" instead.

line 103: "two-dimensional sources" -> I would describe them as two independent sources. I naturally thought about images (2 dimensions), but that's not what's meant. Later in the paper, it becomes clear, but at line 103 it is rather confusing as it is the first time the source model is introduced.

line 104: the distribution D needs to be better explained. It also becomes clear later in the paper, but it should be stated more clearly on l104 as well.

line 489: "Such a biomimetic artificial intelligence

that implements the self-organising mechanisms of neuronal networks is expected to be an alternative to conventional learning algorithms relying on back-propagation" -> is this statement suggesting that backprop can be replaced by Hebbian learning? Would the fixed points be identical? In my view, this statement is very important and deserves more attention. Can it be formalized in a theorem?

Line 527: I dont understand why $A = (1,0.75,0.25,0)$. It would be helpful to explain this a bit better.

The authors should better explain the novelty in the current paper compared to earlier papers, especially [22,23].

Based on those comments, I recommend a major review of the paper. The research results are very interesting and innovative, however, some changes need to be made to the manuscript.

REVIEWER COMMENTS

Reviewer #1 (Remarks to the Author):

The ability to train a living neuronal network to do blind source separation is a noteworthy result of this paper, and I believe the experimental work is sound and supports that claim, with caveats below. I am not sure if this is a new result or not, as other similar papers have been published, including by these authors (e.g., Isomura et al. (2015)). I believe the goal of this paper, however, was to help establish the Free Energy Principle as a useful tool with which to predict eventual learning by neuronal networks given initial learning and activity results. I am not well qualified to judge whether this was accomplished successfully, as Bayesian statistics are not my field of expertise.

[Response]

We would like to thank the reviewer for these gracious comments. As described below, we have revised the manuscript to address the issues raised. We hope these revisions are what you and the editor had in mind.

It would be helpful to define your technical terms carefully for those in other fields who may not be familiar with variational Bayes or FEP. The information in Table 1 is a good start. That said, I will offer some suggestions for improvements of this paper, based on the parts I can follow.

[Response]

To clarify the definition of technical terms for those in other fields who may not be familiar with variational Bayes and the free-energy principle, we have added a glossary of expressions in Table 1, which is now cited in the Introduction section as follows:

“According to the free-energy principle, perception, learning, and action—of all biological organisms—can be described as minimising variational free energy, as a tractable proxy for minimising the surprise (i.e., improbability) of sensory inputs [1,2]. By doing so, neuronal (and neural) networks are considered to perform variational Bayesian inference [3]. **Table 1** provides a glossary of technical terms used commonly in the free-energy principle and active inference literature.”

Table 1. Glossary of terms

Expression	Description
Free-energy principle (FEP)	A principle that can be applied to perception, learning, and action in biological organisms. Technically, the FEP is a variational principle of least action that describes action and perception as, effectively, minimising prediction errors.
Variational Bayesian inference	An approximate Bayesian inference scheme that minimises variational free energy as a tractable proxy for—or bound on—surprise. Minimising surprise is equivalent to maximising the evidence for a generative model. In machine learning, variational free energy is known as an evidence bound.
Prior belief	Probabilistic beliefs about unobservable variables or states prior to receiving observations, denoted as $P(\vartheta)$.
(Approximate) Posterior belief	(Approximate) Bayesian belief about unobservable variables or states after receiving observations, denoted as $Q(\vartheta) \approx P(\vartheta o)$.
Likelihood	The likelihood of an observation given unobservable states, denoted as $P(o \vartheta)$.
Generative model	Probabilistic model that expresses how unobservable states generate observations, defined in terms of the likelihood and prior beliefs $P(o, \vartheta) = P(o \vartheta)P(\vartheta)$.
Surprise	The surprisal or self-information, which scores the improbability of an observation under a

	generative model: defined as $-\ln P(o) = -\ln(\int P(o, \vartheta) d\vartheta)$. Here, $P(o)$ is known as the marginal likelihood or model evidence. It is called the marginal likelihood because it marginalises over the unknown causes an observation.
Variational free energy	An upper bound on surprise—or the negative of an evidence lower bound (ELBO)—defined as $F = E_{Q(\vartheta)}[-\ln P(o, \vartheta) + \ln Q(\vartheta)]$, where $E_{Q(\vartheta)}[\cdot]$ denotes the expectation over $Q(\vartheta)$.
Bayesian belief updating	The process of using observations to update a prior belief to a posterior belief. Usually, in biomimetic schemes, belief updating uses variational Bayesian inference, where neuronal dynamics perform a gradient descent on variational free energy.
Partially observable Markov decision process (POMDP)	A generic generative model that expresses unknown causes of observations in terms of discrete state spaces and categorical distributions.

Neurobiologists would appreciate any new tools to help understand what is going on in living neuronal networks. Has this set of experiments told us anything new about how in vitro networks (and presumably, in vivo ones) learn? If so, please make it more clear what those advances are.

[Response]

Thank you for foregrounding this issue. The novelty and utility of the reverse engineering approach are detailed in the first subsection of the Results section. Furthermore, we have added the following paragraphs to the Discussion section:

“The present work has addressed the predictive validity of the free-energy principle at the circuit level by delineating the functional specialisation and segregation in neuronal networks via free energy minimisation. Identifying the characteristic functions of arbitrary neural networks is not straightforward. However, according to the complete class theorem [41–43], any system that minimises a cost function under uncertainty can be viewed as Bayesian inference. In light of this, we showed that any neural network—whose activity and plasticity minimise a common cost function—can be cast as performing (variational) Bayesian inference [16,17]. Crucially, the existence of this equivalence enables the identification of a natural map from neuronal activity to a

unique generative model (i.e., hypothesis about the external milieu), under which a biological system operates. This step is essential to link empirical data—which report the ‘internal’ circuit dynamics (i.e., physiological phenomena)—to the representation of the ‘external’ dynamics (i.e., functions or computations) that the circuit dynamics imply, in terms of variational Bayesian inference. Using this technique, we fitted stimulus-evoked responses of *in vitro* networks—comprising the cortical cells of rat embryos—to a canonical neural network and reverse engineered an POMDP generative model, apt to explain the empirical data. In other words, we were able to explain empirical responses as inferring the causes of stimuli, under an implicit generative or world model.”

“Furthermore, reverse engineering a generative model from observed responses specifies a well-defined synaptic plasticity rule. Using this rule, we showed that the self-organisation of *in vitro* networks follows a gradient descent on variational free energy under the (POMDP) generative model. In short, the virtues of the reverse engineering are that: (i) when provided with empirical responses, it systematically identifies what hypothesis (i.e., generative model) the biological system employs to infer the external milieu. Moreover, (ii) it offers quantitative predictions about the subsequent self-organisation (i.e., learning) of the system that can be tested using data. This provides a useful tool for analysing and predicting electrophysiological and behavioural responses and elucidating the underlying computational and self-organisation principles.”

“Prior beliefs play an important role in making inferences. Combining mathematical analyses with empirical observations revealed that baseline excitability is a circuit-level encoding of prior beliefs about hidden states. The notion that manipulating the state prior (encoded by neuronal excitability) disrupts inference and learning may explain the perceptual deficits produced by drugs that alter neuronal excitability, such as anxiolytics and psychedelics [53]. This may have profound implications for our understanding of how anxiolytics and psychedelics mediate their effects; namely, a direct effect on baseline activity can alter subsequent perceptual learning. Additionally, aberrant prior beliefs are a plausible cause of the hallucinations and delusions that constitute the positive symptoms of schizophrenia [54,55]. This suggests that, in principle, reverse engineering provides a formal avenue for estimating prior beliefs from empirical data—and for modelling the circuit mechanisms of psychiatric disorders (e.g., synaptopathy). Further, reproduction of these phenomena in *in vitro* (and *in vivo*) networks furnishes the opportunity to elucidate the precise pharmacological, electrophysiological, and statistical mechanisms underlying Bayesian inference in the brain.”

The evidence that applying GABAergic blockers and agonists reduces learning should be bolstered with more controls, and especially, to eliminate the possibility that the network has just been disabled unnaturally by these treatments.

[Response]

Thank you for pointing this out. In the revised manuscript, we have clarified that pharmacological treatments did not unnaturally disrupt network dynamics because the neuronal responses and synaptic plasticity were retained even in the presence of bicuculline or diazepam.

The response level increased after treatment with bicuculline and decreased after treatment with diazepam, but sufficient responses were retained. We have described this in the Results and Methods sections as follows:

“Average response levels were higher in bicuculline-treated cultures than in control cultures. Conversely, diazepam-treated cultures exhibited lower response levels, but retained sufficient responsiveness to analyse response specificity.”

“Out of these samples, 7 cultures in the control condition, 6 treated with bicuculline, and 7 with diazepam were obtained from newly conducted experiments, where their response intensities were 3.0 ± 1.1 , 3.7 ± 1.9 , and 2.3 ± 0.86 spike/trial, respectively (mean \pm standard deviation).”

Furthermore, when changes in effective synaptic connectivity were estimated, a certain amount of plasticity occurred even in the presence of bicuculline or diazepam. We have created an Extended Fig. 1b and have cited it in the Results section as follows:

“Effective synaptic connectivity analysis suggested that a certain amount of plasticity occurred even in the presence of bicuculline or diazepam (**Extended Fig. 1b**).”

Extended Fig. 1. “b Amount of changes in effective synaptic connectivity, indicating the occurrence of synaptic plasticity during the training period. Distributions obtained from $n = 448$, 1920, and 384 connections are shown for diazepam, control, and bicuculline conditions, respectively. The amount of plasticity is characterised by the connectivity change over the training period, defined as $\sum_{k=2}^{100} |W_{ij}^{(k)} - W_{ij}^{(k-1)}|$, where $W^{(k)} = \{W_{ij}^{(k)}\}$ is a matrix of the effective synaptic connectivity at session k and $|\cdot|$ denotes the absolute value. Horizontal bars report the mean.”

In several places (e.g., line 427), reference is made to single-cell or synaptic changes, while the experimental methods described do not address any individual synapses, let alone single cells. I suggest you sort spikes and follow the excitability of sorted units (individual neurons) recorded and stimulated. To get the ground truth of whether a synapse is strengthened or weakened, one must carry out intracellular recordings or perhaps calcium or voltage imaging at the single-cell level. I suggest you refer to changes you have observed in network excitability as a network phenomenon, and steer clear of making any claims about individual cells or synapses unless you do those types of single-cell experiments. I believe your data clearly show you can alter responses of certain (unspecified) circuits within the network, to allow blind source separation.

[Response]

Thank you for pointing this out. We did not mean to imply systematic changes at the single neuron or synapse level. We have now tried to make this clearer. Interestingly, the free-energy principle only applies at the ensemble average level. In more detail, the goal of this work is to show the predictive validity of the free-energy principle. To this end, we focus on ensembles of neurons that exhibit similar functional properties and effective synaptic connectivity from sensory inputs to these ensembles, because neurons are considered to encode information in terms of ensemble averages, i.e., population coding. As per your inputs and suggestions from reviewer #2 (comment #5), we have removed references to single neuron or synapse and revised the manuscript accordingly.

In the Discussion section (line 427 in original manuscript):

“The present work has addressed the predictive validity of the free-energy principle at the circuit level by delineating the functional specialisation and segregation in neuronal networks via free energy minimisation.”

In the Results section and Fig. 2 legend:

“The recorded neuronal responses were categorised into source 1- and source 2-preferring and no-preference groups, depending on the average response intensity, conditioned upon the hidden source (**Fig. 2b**). Note that each electrode can record spiking responses from one or more neurons.”

“This disruption of learning was observed both for source 1- and 2-preferring neuronal responses.”

“**b** Recorded neuronal responses were categorised into source 1-preferring (red), source 2-preferring (blue), and no-preference (grey) groups.”

Presumably, those are due to changes in synaptic strengths, but you have no data to address what is going on at individual synapses, and many alternative possible mechanisms exist for changing

the response properties of a neuronal circuit. It would be helpful if you were to differentiate (by adding several synaptic blockers, e.g.) between post-synaptic responses to stimulation and directly evoked action potentials in the stimulated cells. Contrary to what you claimed on line 699, even the direct responses are prone to vary over time.

[Response]

Thank you for pointing this out. The directly evoked action potentials can be distinguished from subsequent synaptically induced responses based on their high reliability of occurrence (> 80%), low jitter (160 μ s), and consistency of waveform across trials (Bakkum et al., PLoS ONE, 2008). We have confirmed that the properties of spiking responses 10–30 ms after stimulation, that we used for analyses, were more consistent with those of synaptically induced responses. Moreover, Figure 1D in (Bakkum et al., PLoS ONE, 2008) shows that approximately 94% of directly evoked action potentials occur within 10 ms after stimulation. Thus, we think that the majority of spiking responses we analysed were synaptically induced responses, while a few directly evoked action potentials might be involved. Additionally, we have confirmed that changes in response intensities 10–30 ms after stimulation were inhibited by NMDA receptor blocker APV to a certain degree (Fig. 2d), indicating the contribution of synapses. Given your observations, we have revised the sentences in the Results section as follows:

“Response intensity was defined as the number of spikes 10–30 ms after a stimulation (**Fig. 2a**) following the previous treatment [22] (see **Extended Fig. 1a** for other electrodes). This is because a large number of spikes—induced by synaptic input—were observed during that period, while most directly evoked action potentials (which were not the subject of our analyses) occur within 10 ms after stimulation [40].”

Having said this, we agree with your comment “even the direct responses are prone to vary over time” because their latency varies over the range of a few hundred microseconds (Figures 3 and 4 in Bakkum et al., PLoS ONE, 2008) although it is sufficiently shorter than the 20 ms time window that we used for analyses. However, the number of directly evoked action potentials is supposed to be invariant. Thus, we have revised the sentence in the Methods section (line 699 in original manuscript) as follows:

“To identify functionally specialised neurons, we modelled recorded activity as a mixture of the response directly triggered by the stimulus and functionally specialised response to the sources. Most directly triggered responses occur within 10 ms of stimulation, and their number is largely invariant over time, while their latency varies in the range of a few hundred microseconds [40]. Conversely, functionally specialised responses emerge during training, and the majority occur 10–30 ms after stimulation. Thus, analysing the deviation of the number of spikes in this period enables the decomposition of the responses into stimulus- and source-specific components.”

In lines 252-254, it is mistakenly said that GABAergic inputs are upregulated by a GABA antagonist and downregulated by a GABA agonist. This is backwards from what these drugs do. I think you meant to say "network excitability" rather than GABAergic inputs.

[Response]

Thank you for spotting this. We have revised the sentence in the Results section as follows:

“Pharmacological downregulation of gamma-aminobutyric acid (GABA)-ergic inputs (using a GABA_A-receptor antagonist, bicuculline) or its upregulation (using a benzodiazepine receptor agonist, diazepam) altered the baseline excitability of *neuronal networks*.”

You have overstated (e.g., lines 230, 271, 280) the effects of these drugs (and APV), when your data show they reduce the learning effects, while some learning persists.

[Response]

Thank you for pointing this out. We have revised the sentences (lines 230, 271, 280 in original manuscript) to change their nuances as follows:

“These changes were inhibited by *N*-methyl-D-aspartate (NMDA) receptor antagonist, 2-amino-5-phosphonopentanoic acid (APV) to a certain degree (**Fig. 2d**), indicating that the observed self-organisation depends on NMDA-receptor-dependent plasticity.”

“The key notion here is that this simple manipulation was sufficient to account for the failure of inference and subsequent learning, as evidenced by the absence of functional specialisation.”

“In either case, *in vitro* and *in silico* networks failed to perform causal inference, supporting our claim that the failure can be attributed to a biased state prior, under which they operated.”

It would be good if you could separate the performance of the network, while under the influence of blockers and agonists, from their effects on learning by only testing their effects on learning after the drugs have been washed out. Otherwise, you have merely shown that you can disrupt network activity, not learning, which almost any chemical can do.

[Response]

Thank you for this suggestion. However, we have confirmed that learning was inhibited by bicuculline and diazepam. Please note that Figs. 2g, h and 3d and Extended Fig. 2 show the drug effects on plastic changes in neuronal responses and effective synaptic connectivity (i.e., learning) from the initial session—not the direct drug influence on network activity—because the drug was

added before the training period. For further clarification, we have created Extended Fig. 1c to show how plasticity changed and cited it in the Results section, as follows:

“These substances were added to the culture medium before the training period and were therefore present over training.”

“The difference was observed in the specificity of connectivity emerging during the training period (**Extended Fig. 1c**). Here, the specificity was characterised with a gap in the contribution of a sensory electrode to sources 1- and 2-preferring units. While the specificity increased in all groups, it was significantly inhibited in the presence of bicuculline or diazepam.”

Extended Fig. 1. “c Changes in the specificity of effective synaptic connectivity. Here, the specificity is defined as $W_{1j} - W_{2j}$ for $j = 1, \dots, 16$ and $W_{2j} - W_{1j}$ for $j = 17, \dots, 32$. This computes how much the contribution of a sensory electrode differs between sources 1- and 2-preferring ensembles. Please refer to Methods for the definition of W .”

Furthermore, we do not claim that the changes in the prior belief about hidden states—encoded by the firing threshold (baseline excitability)—only affect the learning. Our mathematical analyses have shown that changing prior beliefs alters both inference and subsequent learning (Isomura & Friston, 2020). The effect of bicuculline or diazepam disabling *in vitro* neurons to selectively respond to sources (Fig. 2g, h) was consistent with the expected effect of the biased prior belief (Fig. 2e, f), in terms of both inference and learning. In short, the free-energy principle suggests that the alteration of network activity that “almost any chemical can do” is sufficient to change the prior belief and subsequent learning. We have discussed this in the Discussion section as follows:

“These empirical data and complementary modelling results also explain the strong influence of prior beliefs on perception and causal inference—and the disruptive effects of drugs on perception in neuronal networks. Both synaptic plasticity and inference depend on convergent neuronal activity; therefore, aberrant inference will disrupt learning. Conversely, inference is not possible without the knowledge accumulated through experience (i.e., learning). Thus, inference is strongly linked to learning about contingencies that generate false inferences. Our findings demonstrate this association both mechanistically and mathematically, in terms of one simple rule that allows

prior beliefs to underwrite inferences about hidden states.”

As for the washout of drugs, this corresponds to changing the prior belief after learning has been established. However, such a situation is beyond the scope of this work. This work focused on settings in which prior beliefs were fixed. Examining biologically plausible Bayesian models for a situation with dynamically changing priors is more challenging than the current agenda—and is an interesting target for future work along these lines.

Reviewer #2 (Remarks to the Author):

The paper describes a thorough experimental validation of the free-energy principle for quantitative prediction of neuronal network responses of in-vitro neuronal cultures and in-silico neural network simulations. Pharmacology is used to show that changes in synaptic connectivity affects variational free energy.

The work builds on the authors former publications (ref 20-23), and here significantly extends it, by showing an experimental validation of the quantitative prediction of self-organization and trajectory of plasticity of in-vitro neuronal networks as measured with microelectrode arrays. Especially interesting and novel is the pharmacological manipulations demonstrating that the effects of compounds on excitability are very well aligned between the in-vitro model and the in-silico simulations using this method.

The method is well described and thorough, providing sufficient information to reproduce it by other groups. I recommend the work to be published with minor corrections, as listed below:

[Response]

We would like to thank the reviewer for these valuable comments. As described below, we have revised the manuscript to address the issues raised. We hope these revisions are what you and the editor had in mind.

- L212: provide more details on how these stimuli were “carefully constructed”

[Response]

Thank you for pointing this out. The detailed construction is introduced in the preceding section and is shown in Fig. 1a. We have also changed the sentence in the Results section as follows:

“Neurons were stimulated with the above-constructed patterns of sensory inputs (see the

preceding section), comprising 32 binary sensory inputs (o) that were generated from two sequences of independent binary hidden sources (s) in the manner of the POMDP generative model above (**Fig. 1a**, right)."

- L219: for clarity, specify explicitly that each of the 100 session was an identical repetition of the "random" sequence within each 256s long session.

[Response]

No problem. We have added the following sentence to the Results section:

"The training comprised 100 sessions, each of which was an identical repetition of the 256 s-long random sequence."

- L222: please provide some justification on the selected repones window of 10-30ms (or reference older work where this was justified).

[Response]

Thank you for suggesting this. We have revised this sentence in the Results section as follows:

"Response intensity was defined as the number of spikes 10–30 ms after a stimulation (**Fig. 2a**) following the previous treatment [22] (see **Extended Fig. 1a** for other electrodes). This is because a large number of spikes—induced by synaptic input—were observed during that period, while most directly evoked action potentials (which were not the subject of our analyses) occur within 10 ms after stimulation [40]."

- L225: Additional data showing how stable this "classification" into the two groups was would be very interesting. Was this classification dependent on the carefully constructed o ? Did it change over time? Any spatial correlation to o ? Was it for all conditions roughly an equal number of electrodes in each group, and it not, was this somehow accounted for?

[Response]

Thank you for highlighting this issue. The following clarifications are provided for each question.

As for the question "Was this classification dependent on the carefully constructed o ?", we have added the following sentences to the Methods section 'Data preprocessing':

"Sources 1- and 2-preferring ensembles were quantitatively similar because the total contribution from sources 1 and 2 to stimuli o_t was designed to be equivalent, owing to the symmetric structure of the A matrix. Under this setting, this similarity was conserved, irrespective

of the details of the mixing matrix A .”

As for the question “Did it change over time?”, the time variability was small. Thus, we adopted all session averages to compute the mean response intensity used for classification.

As for the question “Any spatial correlation to o ?”, stimuli o were introduced from randomly selected electrodes, so there was no spatial correlation. We have added the following sentence to the Results section:

“The 32 stimulation electrodes were randomly distributed over 8×8 MEAs in advance and fixed over training.”

As for the question “Was it for all conditions roughly an equal number of electrodes in each group, and if not, was this somehow accounted for?”, the neurons in all samples were generally split with a similar balance. We have added a sentence that describes the mean number of neurons \pm standard deviation to the Methods section ‘Data preprocessing’ as follows:

“Note that the number of source 1-preferring, source 2-preferring, and no preference electrodes in each sample are 17.1 ± 7.0 , 15.0 ± 7.0 , and 11.5 ± 6.7 , respectively ($n = 30$ samples under the control condition).”

• L223: Correct to assume no spike sorting was performed, and “neurons” here actually correspond to “electrode”, with spikes from potentially multiple neurons mixed together. If so, please clarify in the text. Same also e.g. on L233, and other places in the MS.

[Response]

Thank you for highlighting this. We have revised the sentences in the Results section (including line 223 of original manuscript) and Fig. 2 legend as follows:

“The recorded neuronal responses were categorised into source 1- and source 2-preferring and no-preference groups, depending on the average response intensity, conditioned upon the hidden source (**Fig. 2b**). Note that each electrode can record spiking responses from one or more neurons.”

“This disruption of learning was observed both for source 1- and 2-preferring neuronal responses.”

“**b** Recorded neuronal responses were categorised into source 1-preferring (red), source 2-preferring (blue), and no-preference (grey) groups.”

Additionally, we have replaced the term ‘at a single-neuron level’ on line 233 (original manuscript) and other places with ‘at a cellular level’ to describe the results more accurately as follows:

“These results indicate the occurrence of blind source separation at a cellular level—through activity-dependent synaptic plasticity—supporting the theoretical notion that neural activity encodes the posterior belief (i.e., expectation) about hidden sources or states [1,2].”

“We observed that both hyper-excitability (**Fig. 2g**, right) and hypo-excitability (**Fig. 2g**, left) significantly suppressed the emergence of response specificity at the cellular level (**Fig. 2h**).”

- Fig2c: y-axis unit is likely not “spike/trial”, but unit-less, as it is showing the “change”. Correct? Please correct, also in all other similar figures.

[Response]

The y-axis in Fig. 2c shows the change (i.e., deviation) from the baseline response level with the unit of spike/trial. The deviation from the initial state has the same units as the initial state. Thus, the y-axis in Fig. 2c also has the units ‘spike/trial’.

- Fig2a: correct that the white raster plot for $s=(0,0)$ is showing spontaneous activity? Why is there not even a single spike in this section?

[Response]

Thank you. Yes, the white raster plot for $s = (0,0)$ in Fig. 2a corresponds to spontaneous activity without stimulation. No activity was observed at this electrode during this period by chance. A certain level of spontaneous activity was recorded at the other electrodes, while its intensity was lower than that of the stimulus-evoked response. We have created an Extended Fig. 1a to show the early evoked responses recorded at other electrodes and have cited it in the Results section as follows:

“Response intensity was defined as the number of spikes 10–30 ms after a stimulation (**Fig. 2a**) following the previous treatment [22] (see **Extended Fig. 1a** for other electrodes).”

“**Extended Fig. 1. Supplementary data. a** Early evoked responses of *in vitro* neurons recorded at a single electrode. Two examples are shown.”

- **Method Cell Culture:** was there a primary or secondary coating used? Please specify.

[Response]

No problem. The procedure followed the treatment described in previous work [22]. We have added the following information to the Methods section ‘Cell culture’:

“Half a million dissociated cortical cells (a mixture of neurons and glial cells) were seeded on the centre of MEA dishes, where the surface of MEA was previously coated with polyethyleneimine (Sigma–Aldrich, St. Louis, MO, USA) overnight.”

Reviewer #3 (Remarks to the Author):

In earlier work, the authors show the connection between cost minimization in neural networks on the one hand and minimizing variational free energy on the other hand. This relationship enables the interpretation of learning in neural networks as variational inference.

In the current paper, the authors extend this work to show that variational free energy minimisation can quantitatively *predict* the self-organisation of neuronal networks in terms of their responses and plasticity.

[Response]

We would like to thank the reviewer for these nice comments. As described below, we have

revised the manuscript to address the issues raised. We hope these revisions are what you and the editor had in mind.

Major comments:

1. It is probably fair to say that predicting the plasticity and responses in the biological neural networks can be achieved by simulating the corresponding artificial neural networks, by implementing (6) and Hebbian dynamics for the weight updates. Therefore, it is not the case that the variational inference approach is the only way to predict the synaptic weights and responses. However, it is indeed a nice interpretation of the latter in terms of inference. But I would argue that neural computations have been interpreted as Bayesian inference before by numerous researchers. What is precisely the novelty in this paper? Is the fact that one can do numerical predictions and interpret the variables in the neural net in terms of random variables that are being inferred?

[Response]

Yes, it is exactly the last observation; namely, one can identify the latent states that are being inferred. This is very similar to being able to explain and interpret neural networks in machine learning; in terms of an implicit generative or world model. However, we do not claim that this method is the only way, but unlike conventional methods, we had previously shown that the dynamics of canonical neural networks are mathematically equivalent to a particular variational Bayesian inference (Isomura & Friston, 2020; Isomura et al., 2022). This proof underwrites the uniqueness and novelty of the proposed approach. As per your inputs and suggestions from reviewer #1 (comment #2), we have added material that explains the novelty of this work in the Discussion section (please see above). Further, we have added the following paragraph to the Discussion section:

“Although numerous neural implementations of Bayesian inference have been proposed [44–46], these approaches generally derive update rules from Bayesian cost functions without establishing the precise relationship between these update rules and the neural activity and plasticity of canonical neural networks. The reverse engineering approach differs conceptually by asking what Bayesian scheme could account for any given neuronal dynamics or neural network. By identifying the implicit inversion scheme—and requisite generative model—one can then lend any given network an interpretability and explainability. In the current application of this approach, we first consider a biologically plausible cost function for neural networks that explains both neural activity and synaptic plasticity. We then identify a particular generative model under which variational free energy is equivalent to the neural network cost function. In this regard, the reverse engineering offers an objective procedure for explaining neural networks in terms of Bayesian inference.”

As for the necessity of a variational Bayesian inference perspective, if one ignores it and discuss only the dynamics of neural networks, one cannot explain under what plasticity conditions (and why) neural networks can infer hidden sources. The virtue of the free-energy principle is that it lends an explainability to neuronal network dynamics and architectures, in terms of variational Bayesian inference. We discuss this in the first subsection of the Results section.

Of course, the inference problem at hand is rather simple, therefore, there might not be much benefit from viewing the computations as inference here.

[Response]

While this is true, one could argue that the problem is not so simple. Hebbian dynamics is a broad term that does not commit to a particular functional form. The plasticity formula must be a carefully constructed combination of Hebbian and homeostatic plasticity to perform inferences appropriately. The free-energy principle explains why this is the case and identifies an apt Hebbian rule. To make this argument, we have added the following paragraph to the Discussion section:

“Further, the synaptic plasticity rule is derived as the gradient descent on the cost function that is determined by the integral of neural dynamics. Crucially, learning through this plasticity rule can be read, formally, as Bayesian belief updating under an appropriate generative model. Conversely, naive Hebbian plasticity rules—with an *ad hoc* functional form—correspond to Bayesian belief updating under a suboptimal generative model with biased prior beliefs, which cannot solve simple blind source separation problems [16]. As predicted, *in vitro* neural networks above failed to perform blind source separation, with changed baseline excitability and implicit priors. In short, the free-energy principle is necessary to determine the optimal balance between Hebbian and homeostatic plasticity that enables blind source separation by *in vitro* networks.”

2. I think it would be interesting to speculate about contexts where the inference viewpoint is essential to understand the phenomenon. This could be in the context of cognitive psychology or psychiatry, where changing priors leads to different behaviors (potentially erratic ones). Could the authors perhaps extend the inference problem at hand to such scenario? For instance, binocular rivalry, where there also seem to be two "sources" or two competing explanations. Having such illustration/example may better underline the potential of the formalism introduced in the paper, since the considered inference problem is a simple and abstract, and a too distant from realistic inference problems.

[Response]

Thank you for these suggestions. Although we disagree with the criterion on the simplicity and

abstraction of the inference task because the blind source separation and cocktail party effect are complicated and concrete cognitive phenomena (Brown et al., 2001; Mesgarani & Chang, 2012), we totally agree with the importance of prior beliefs in characterising inferences performed by biological organisms. Thus, we have added the following paragraph to the Discussion section accordingly:

“Prior beliefs play an important role in making inferences. Combining mathematical analyses with empirical observations revealed that baseline excitability is a circuit-level encoding of prior beliefs about hidden states. The notion that manipulating the state prior (encoded by neuronal excitability) disrupts inference and learning may explain the perceptual deficits produced by drugs that alter neuronal excitability, such as anxiolytics and psychedelics [53]. This may have profound implications for our understanding of how anxiolytics and psychedelics mediate their effects; namely, a direct effect on baseline activity can alter subsequent perceptual learning. Additionally, aberrant prior beliefs are a plausible cause of the hallucinations and delusions that constitute the positive symptoms of schizophrenia [54,55]. This suggests that, in principle, reverse engineering provides a formal avenue for estimating prior beliefs from empirical data—and for modelling the circuit mechanisms of psychiatric disorders (e.g., synaptopathy). Further, reproduction of these phenomena in *in vitro* (and *in vivo*) networks furnishes the opportunity to elucidate the precise pharmacological, electrophysiological, and statistical mechanisms underlying Bayesian inference in the brain.”

Moreover, we have added the following paragraph to the Methods section ‘Generative process’:

“The experimental paradigm established in previous work [22] was employed. The blind source separation addressed in this work is an essential ability for biological organisms to identify hidden causes of sensory information, as considered in the cocktail party effect [27,28]. This deals with the separation of mixed sensory inputs into original hidden sources in the absence of supervision, which is a more complex problem than naive pattern separation tasks.”

We note that the binocular rivalry—a phenomenon of visual perception in which different images presented to each eye are perceived alternately—corresponds to a pattern separation task that does not deal with the separation of mixed sensory inputs into original hidden sources.

3. The authors rely on variational inference. Is it possible that other inference mechanisms would to similar results and interpretations, e.g., Monte Carlo or belief propagation? I don't understand why the mean field approach is essential. Or is it essential? Could also more complex mean field methods (structured mean field) be mapped to neural computations?

[Response]

Thank you for raising these interesting points. We did not assume (rely on) variational Bayesian inference, but derived variational Bayesian inference—under a particular mean-field approach—from a biologically plausible canonical neural network model (Isomura & Friston, 2020; Isomura et al., 2022). In contrast, Monte Carlo and belief propagation differ from canonical neural networks. Whether or not they are equivalent to some biologically plausible neural network models has yet to be established. Thus, they were outside the scope of this work. We have added the following to the Methods section ‘Canonical neural networks’ to describe why the mean-field approach is essential to model neural networks as follows:

“Further, reverse-engineering can naturally derive variational Bayesian inference—under a particular mean-field approximation defined in equation 2—from a canonical neural network architecture. This representation of posterior beliefs is essential for the networks to encode rapidly changing hidden states (s_τ) and slow parameters (A) with neural activity (x) and synaptic strengths (W), respectively. In this setting, a mean field approximation implements a kind of adiabatic approximation, in which the separation of timescales between fast neuronal responses and slow learning is leveraged to increase the efficiency of inference. Please see ref. [16] for further discussion.”

Regarding the question “Could also more complex mean field methods (structured mean field) be mapped to neural computations?,” perhaps other types of neural network models may be equivalent to other types of structured mean-field models, as discussed in the previous paper (Isomura & Friston, 2020), but examples of such network models are non-trivial.

We have not expanded on this interesting issue; however, there are lots of things that one could talk about. For example, one could compare generative models of discrete states (as in the current example) with models of continuous states that would speak to predictive coding and Kalman filtering (and possibly particle filtering). The question about structured mean field assumptions is also interesting and would speak to hierarchical generative models (not dissimilar to hierarchical Dirichlet processes). Belief propagation and variational message passing (with variational or Bethe free energies) are further interesting variations; however, much of this work would be difficult to connect to neuronal dynamics *per se* because it is based upon fixed point iteration (which does not generate continuous time series of the sort required to explain neuronal message passing). We will think more about this.

4. The paper only deals with *in vitro* biological neural networks. Can the conclusions be generalized to *in vivo* neural networks? Can similar experiments be conducted *in vivo*? What would be the challenges? I recommend the authors to reflect on this topic in the Discussion section.

[Response]

Thank you for highlighting this issue. This is exactly what we hope to do in our future work. We have added the following revised paragraph to the Discussion section accordingly.

“Importantly, although this paper focused on a comparison of *in vitro* data and theoretical prediction, the reverse engineering approach is applicable to characterising *in vivo* neuronal networks, in terms of their implicit generative model with prior beliefs. It can, in principle, be combined with electrophysiological, functional imaging, and behavioural data—and give predictions, if the learning process is continuously measured. Thus, the proposed approach for validating the free-energy principle can be applied to the neural activity data from any experiment that entails learning or self-organisation; irrespective of the species, brain region, task, or measurement technique. Even in the absence of learning, it can be applied, if one can make some theoretical predictions and compare them with experimental data. For accurate predictions, large-scale and continuous measurements of activity data at the population level from pre-learning to post-learning stages would be a prerequisite. In future work, we hope to test, empirically, whether the free-energy principle can qualitatively predict the perception, learning, and behaviour of various biological systems.”

Minor comments:

line 84: "as neuronal networks and reserve the term neural network for an *in silico* model" -> I would suggest the terminology "biological neural networks (BNN)" and "artificial neural networks (ANN)" instead.

[Response]

Thank you for your suggestion. However, the term ‘artificial neural network’ is a broad term, including less biological plausible ones used in machine learning. In contrast, the canonical neural network considered in this work has been derived from realistic neuron models, such as the Hodgkin–Huxley model (Isomura et al., 2022). Thus, we believe that the canonical neural network is a plausible computational model of biological neural networks, as proven in this work. From this perspective, the use of the terms ‘biological neural networks’ and ‘artificial neural networks’ is not in line with our intentions. Therefore, in this paper, we decided to use the terminology ‘neuronal networks’ for *in vitro* networks and ‘neural networks’ for canonical neural network models.

line 103: "two-dimensional sources" -> I would describe them as two independent sources. I naturally thought about images (2 dimensions), but that's not what's meant. Later in the paper, it becomes clear, but at line 103 it is rather confusing as it is the first time the source model is introduced.

[Response]

Thank you for pointing this out. As per your suggestion, we have replaced ‘two-dimensional binary hidden sources’ with ‘two binary hidden sources’.

line 104: the distribution D needs to be better explained. It also becomes clear later in the paper, but it should be stated more clearly on l104 as well.

[Response]

Thank you for your suggestion. D is the prior belief about hidden states in the form of a categorical distribution. We have revised the sentence as follows:

“Here, two binary hidden sources $s = (s_1, s_2)$ were sampled at random from a prior categorical distribution $D = (D_1, D_0)$ in a mutually independent manner, where D_1 and D_0 are posterior expectations that satisfy $D_1 + D_0 = 1$. Then, 32 sensory inputs $o = (o_1, \dots, o_{32})$ were generated from s with a categorical distribution characterised by a mixing matrix A .”

line 489: "Such a biomimetic artificial intelligence that implements the self-organising mechanisms of neuronal networks is expected to be an alternative to conventional learning algorithms relying on back-propagation" -> is this statement suggesting that backprop can be replaced by Hebbian learning? Would the fixed points be identical? In my view, this statement is very important and deserves more attention. Can it be formalized in a theorem?

[Response]

Thank you for rising these issues. Various alternative back-propagation methods have been proposed—some recent examples include the forward-forward algorithm proposed by Hinton (2022). Having said this, to the best of our knowledge, whether they have fixed points and structural stability identical to the original back-propagation has not yet been proven. There is also some interesting work from the Oxford and Sussex groups comparing predictive coding with back propagation. This is relevant because predictive coding can be formulated in terms of variational free energy minimisation.

From our perspective, we hope to develop a scheme to transform a wide range of learning algorithms into local learning rules based on the equivalence established in the previous work. Accordingly, we have added the following revised sentence to the Discussion section:

“The generic mechanisms for acquiring generative models can be used to construct a neuromorphic hardware for universal applications [56,57]. Back-propagation is central in many current deep learning methods, but biologically implausible. This has led to various biologically plausible alternatives; e.g., [58–61], some of which appeal to predictive coding formulations of

variational free energy minimisation. The equivalence between neural networks and variational Bayes could be useful to establish biologically plausible learning algorithms, because Hebbian learning rules derived under this scheme are local (energy-based) algorithms. This is because the implicit cost function (contribution to variational free energy as an extensive quantity) can be evaluated locally. Such a biomimetic artificial intelligence—that implements the self-organising mechanisms of neuronal networks—could offer an alternative to conventional learning algorithms such as back-propagation, and to have the high data, computational, and energy efficiency of biological computation [62,63]. This makes it promising for the next-generation of artificial intelligence. In addition, the creation of biomimetic artificial intelligence may further our understanding of the brain.”

Line 527: I don't understand why $A = (1,0.75,0.25,0)$. It would be helpful to explain this a bit better.

[Response]

Thank you for rising this point. Accordingly, we have revised the sentence in the Methods section as follows:

“Half of the electrodes ($1 \leq i \leq 16$) conveyed the source 1 signal with a 75% probability or the source 2 signal with a 25% probability. Because each element of the A matrix represents the conditional probability that o_τ occurs given $s_\tau = (s_\tau^{(1)}, s_\tau^{(2)})$, this characteristic is expressed as $A_{1..}^{(i)} = \left(P(o_\tau^{(i)} = 1 | s_\tau = (1,1)), P(o_\tau^{(i)} = 1 | s_\tau = (1,0)), P(o_\tau^{(i)} = 1 | s_\tau = (0,1)), P(o_\tau^{(i)} = 1 | s_\tau = (0,0)) \right) = (1,0.75,0.25,0)$.”

The authors should better explain the novelty in the current paper compared to earlier papers, especially [22,23].

[Response]

Thank you for bringing this to our notice. As per your inputs and suggestions from reviewer #1 (comment #2), we have added paragraphs that explain the novelty of this work under the Discussion section. Further, we have added the following paragraph to the Methods section ‘Data prediction using the free-energy principle’:

“We note that the reverse engineering approach provides three novel aspects compared to earlier work [22,23]. First, previous work assumed the form of the generative model and did not examine whether all elements of the generative model corresponded to biological entities at the circuit level. In the present work, we objectively reverse engineered the generative model from

empirical data and showed a one-to-one mapping between all the elements of the generative model and neural network entities. Second, previous work did not examine the impact of changing prior beliefs on Bayesian inference performed by *in vitro* neural networks. The present work analysed how Bayesian inference and free energy reduction changed when the prior belief and external environment were artificially manipulated and showed that the results were consistent with theoretical predictions. This work validated the predicted relationship between baseline excitability and prior beliefs about hidden states. Third, previous work did not investigate whether the free-energy principle can qualitatively predict the learning process of biological neural networks based exclusively on initial empirical data. This was demonstrated in the current work.”

Based on those comments, I recommend a major review of the paper. The research results are very interesting and innovative, however, some changes need to be made to the manuscript.

[Response]

Thank you for your valuable inputs. We hope these responses and the revised manuscript are satisfactory.

References

16. Isomura, T. & Friston, K. J. Reverse-engineering neural networks to characterize their cost functions. *Neural Comput.* **32**, 2085–2121 (2020). 10.1162/neco_a_01315, Pubmed:32946704.
17. Isomura, T., Shimazaki, H. & Friston, K. J. Canonical neural networks perform active inference. *Commun. Biol.* **5**, 55, (2022). 10.1038/s42003-021-02994-2, Pubmed:35031656.
22. Isomura, T., Kotani, K. & Jimbo, Y. Cultured cortical neurons can perform blind source separation according to the free-energy principle. *PLoS Comput. Biol.* **11**, e1004643 (2015). 10.1371/journal.pcbi.1004643, Pubmed:26690814.
23. Isomura, T. & Friston, K. J. In vitro neural networks minimise variational free energy. *Sci. Rep.* **8**, 16926 (2018). 10.1038/s41598-018-35221-w, Pubmed:30446766.
27. Brown, G. D., Yamada, S. & Sejnowski, T. J. Independent component analysis at the neural cocktail party. *Trends Neurosci.* **24**, 54–63 (2001). 10.1016/s0166-2236(00)01683-0, Pubmed:11163888.
28. Mesgarani, N. & Chang, E. F. Selective cortical representation of attended speaker in multi-talker speech perception. *Nature* **485**, 233–236 (2012). 10.1038/nature11020, Pubmed:22522927.

40. Bakkum, D. J., Chao, Z. C. & Potter, S. M. Long-term activity-dependent plasticity of action potential propagation delay and amplitude in cortical networks. *PLoS ONE* **3**, e2088 (2008). 10.1371/journal.pone.0002088, Pubmed:18461127.
54. Fletcher, P. C. & Frith, C. D. Perceiving is believing: a Bayesian approach to explaining the positive symptoms of schizophrenia. *Nat. Rev. Neurosci.* **10**, 48–58 (2009). 10.1038/nrn2536, Pubmed:19050712.
55. Friston, K. J., Stephan, K. E., Montague, R. & Dolan, R. J. Computational psychiatry: the brain as a phantastic organ. *Lancet Psychiatry* **1**, 148–158 (2014). 10.1016/S2215-0366(14)70275-5, Pubmed:26360579.
58. Schiess, M., Urbanczik, R. & Senn, W. Somato-dendritic synaptic plasticity and error-backpropagation in active dendrites. *PLoS Comput. Biol.* **12**, e1004638 (2016). 10.1371/journal.pcbi.1004638, Pubmed:26841235.
61. Hinton, G. The forward-forward algorithm: Some preliminary investigations. *arXiv* arXiv:2212.13345 (2022). <https://arxiv.org/abs/2212.13345>

REVIEWERS' COMMENTS

Reviewer #1 (Remarks to the Author):

You have revised the paper to my satisfaction, improving it greatly. Thank you for your careful edits, changes, and clarifications.

Reviewer #2 (Remarks to the Author):

Thank you very much for extensively addressing all questions in detail. All concerns raised have been answered sufficiently. I recommend to publish the article without further changes.

Reviewer #3 (Remarks to the Author):

The authors have addressed my earlier suggestions in an adequate manner. Therefore, I recommend this paper to be published.